# Senescent tumor cells lead the collective invasion in thyroid cancer

Young Hwa Kim[1], Yong Won Choi[1,2], Jeonghun Lee[3], Euy Young Soh[3], Jang-Hee Kim[4] & Tae Jun Park[1]

Cellular senescence has been perceived as a barrier against carcinogenesis. However, the senescence-associated secretory phenotype (SASP) of senescent cells can promote tumorigenesis. Here, we show senescent tumour cells are frequently present in the front region of collective invasion of papillary thyroid carcinoma (PTC), as well as lymphatic channels and metastatic foci of lymph nodes. In *in vitro* invasion analysis, senescent tumour cells exhibit high invasion ability as compared with non-senescent tumour cells through SASP expression. Collective invasion in PTC is led by senescent tumour cells characterized by generation of a C-X-C-motif ligand (CXCL)12 chemokine gradient in the front region. Furthermore, senescent cells increase the survival of cancer cells via CXCL12/CXCR4 signalling. An orthotopic xenograft *in vivo* model also shows higher lymphatic vessels involvement in the group co-transplanted with senescent cells and cancer cells. These findings suggest that senescent cells are actively involved in the collective invasion and metastasis of PTC.

[1] Department of Biochemistry and Molecular Biology, Ajou University, School of Medicine, Suwon 443-721, Korea. [2] Department of Hemato-oncology, Ajou University School of Medicine, Suwon 443-721, Korea. [3] Department of Surgery, Ajou University School of Medicine, Suwon 443-721, Korea. [4] Department of Pathology, Ajou University School of Medicine, Suwon 443-721, Korea. Correspondence and requests for materials should be addressed to J.-H.K. (email: drjhk@ajou.ac.kr) or to T.J.P. (email: park64@ajou.ac.kr).

Invasion and metastasis are hallmarks of cancer[1,2]. Invasion is a critical step in the progression to metastasis. For invasion, tumour cells modify not only their shape, but also their attachment to other cells and to the extracellular matrix (ECM). This alteration is known as the 'epithelial–mesenchymal transition' (EMT) and is characterized by loss of cell to cell adhesion molecules (E-cadherin) and upregulated expression of adhesion molecules associated with cell migration (N-cadherin)[3,4]. Through the EMT, tumour cells can detach from the main mass, and the separated tumour cells can invade into the ECM, as well as blood or lymphatic vessels as individual single cell. Therefore, the EMT is supposed to be involved in most steps of tumour progression, from invasion to metastasis, by conferring the abilities to invade, resist apoptosis and disseminate to tumour cells[1]. However, the underlying mechanism of invasion and metastasis varies depending on the type of cancer. Although certain types of high-grade and mesenchymal tumours infiltrate by single-cell migration with EMT characteristics, most low-grade tumours retain cell-to-cell adhesions and invade as cohesive multicellular strands. This type of invasion is known as 'collective invasion.' In carcinomas, originating from breast, colon, prostate and the thyroid gland, cancer cells invade cohesively with features of collective invasion[5].

In collective invasion, most cancers are composed of varying degrees of heterogeneous subpopulations with distinct biologic properties involving proliferative ability, genetic alterations, signal pathways, drug or immune response, angiogenic potential, cell metabolism, motility, secretome and senescence, as well as different abilities for invasion and metastasis; certain cancer cells invade in the front of collective invasion as leaders whereas others are located in the rear and follow[6–8]. Among these biological properties, cellular senescence has been suggested as a barrier against carcinogenesis, because senescence induced by oncogenic activation (oncogene-induced senescence; OIS) is commonly observed in premalignant tumours, but rare in their malignant counterparts[9]. However, recent evidence indicates that cellular senescence can promote carcinogenesis by producing various growth factors, cytokines and proteases, collectively referred to as the senescent-associated secretory phenotype (SASP)[10]. Although senescent cells are rarely observed in cancers, the existence of isolated senescent cells in cancers has also been reported[11–15]. In our previous study involving papillary thyroid carcinoma (PTC), we demonstrated the presence of senescent cells in PTC[16]. Furthermore, our preliminary investigation frequently detected senescence associated-β-galactosidase (SA-β-Gal) positive senescent tumour cells in the invasive borders of PTC, lymphatic channels and metastatic foci of lymph nodes displaying features of collective invasion. These observations led us to hypothesize that senescent cells could participate in PTC invasion and metastasis. To explore this hypothesis, we analysed BRAFV600E-expressing PTC tissues from patients and employed an in vitro senescent thyrocyte model using BRAFV600E oncogenic activation, which is known as the most common oncogenic driver in PTC[17], and applied this in vitro model and an orthotopic xenograft nude mouse model to characterize senescent cells and determine their involvement in collective invasion of PTC.

## Results

**Senescent tumour cells are identified in thyroid cancer**. We examined senescent cells in various tumour types, including thyroid, breast, colon and stomach cancers by SA-β-Gal staining (Supplementary Fig. 1), a standard biomarker of senescence, and found that senescent cells were frequently identified in thyroid cancer, and mostly in BRAFV600E-expressing PTC. To characterize senescent cells in PTC, we performed SA-β-Gal staining using snap-frozen tissue sections from 70 cases of

BRAFV600E-expressing PTC. Interestingly, however, senescent cells were not evenly distributed in the tumours, but more frequently observed at the invasive borders of PTC (Fig. 1a and Supplementary Fig. 2). Since p16INK4A is a classic marker of OIS and have proved useful marker of senescence in vivo, we performed immunostaining for p16INK4A using serial snap-frozen sections of the corresponding PTC and compared the expression of p16INK4A with that of SA-β-Gal. We observed that the intensity and distribution of p16INK4A-immunopositive cells were correlated with those of SA-β-Gal positive cells (Fig. 1b). Real-time polymerase chain reaction (PCR) analysis also indicated an increase in p16INK4A mRNA expression in PTC (Supplementary Fig. 3). Since cellular senescence is characterized by a stable cell cycle arrest, we also performed Ki67 (MIB-1) immunostaining to evaluate the proliferative activity in p16INK4A-immunopositive cells and confirmed the absence of Ki67 (MIB-1) expression in these cells (Fig. 1b). These data together revealed that senescent cells in thyroid cancer were frequently present at the invasive borders. To determine whether p16INK4A immunopositive senescent cells were thyroid tumour cells or stromal cells, we assessed the expression of BRAFV600E using VE1, a BRAFV600E-specific monoclonal antibody and thyroid transcription factor-1 (TTF-1) in formalin-fixed paraffin embedded (FFPE) tissues of BRAFV600E-expressing PTC. We found that p16INK4A expressing senescent cells were also immunopositive for TTF-1 and BRAFV600E, indicating that senescent cells were not stromal cells, but neoplastic epithelial cells (Fig. 1c).

**Senescent cells exist at the collective invasion border**. In PTC invasive regions, neoplastic epithelial cells were organized into multicellular units and exhibited well preserved the E-cadherin expression, indicating collective invasion (Fig. 2a). Furthermore, collective invasion was maintained through lymph node metastasis from the primary cancer lesion. The expression of E-cadherin was preserved not only in primary PTC, but also in the metastatic foci of lymph nodes, and E-cadherin was present at tumour emboli in the D2-40 positive lymphovascular channels (Fig. 2a). Interestingly, senescent tumour cells were frequently located at the front region of E-cadherin conserved collective invasion (Fig. 2b). Since the EMT is known to broadly regulate cancer invasion and metastasis, we next assessed whether the EMT occurred in senescent cells by investigating the expression of EMT markers at the invasive border of PTC where senescent cells were detected. Twist1 and Zeb1 immunohistochemistry was performed using FFPE tissues of PTC, but no increase in the expression of Twist1 and Zeb1 at the invasive border was observed as compared with the corresponding cancer centre. In addition, E-cadherin expression was retained not only at the cancer centre, but also at the invasive border, whereas N-cadherin was hardly detected (Fig. 2c). Taken together, these data revealed that BRAFV600E-expressing senescent tumour cells were present at the invasive borders of PTC with features of collective invasion.

**Senescent tumour cells have invasive ability via SASP**. Since senescent tumour cells are present at the invasive borders of PTC, we next determined whether senescent cells have the ability to invade the matrix. Thus, we isolated tumour cells from BRAFV600E-expressing PTC. The isolated primary tumour cells were composed of both senescent and non-senescent tumour cells (senescent cells comprised 22% of the total). However, the invasion assay showed that 62% of invaded cells were SA-β-Gal positive, whereas only 9.5% of remaining cells (non-invasive cells) in the upper portion of the membrane were SA-β-Gal positive (Fig. 3a). Although the possibility that senescent cells enable non-senescent cells to migrate/invade and the non-senescent

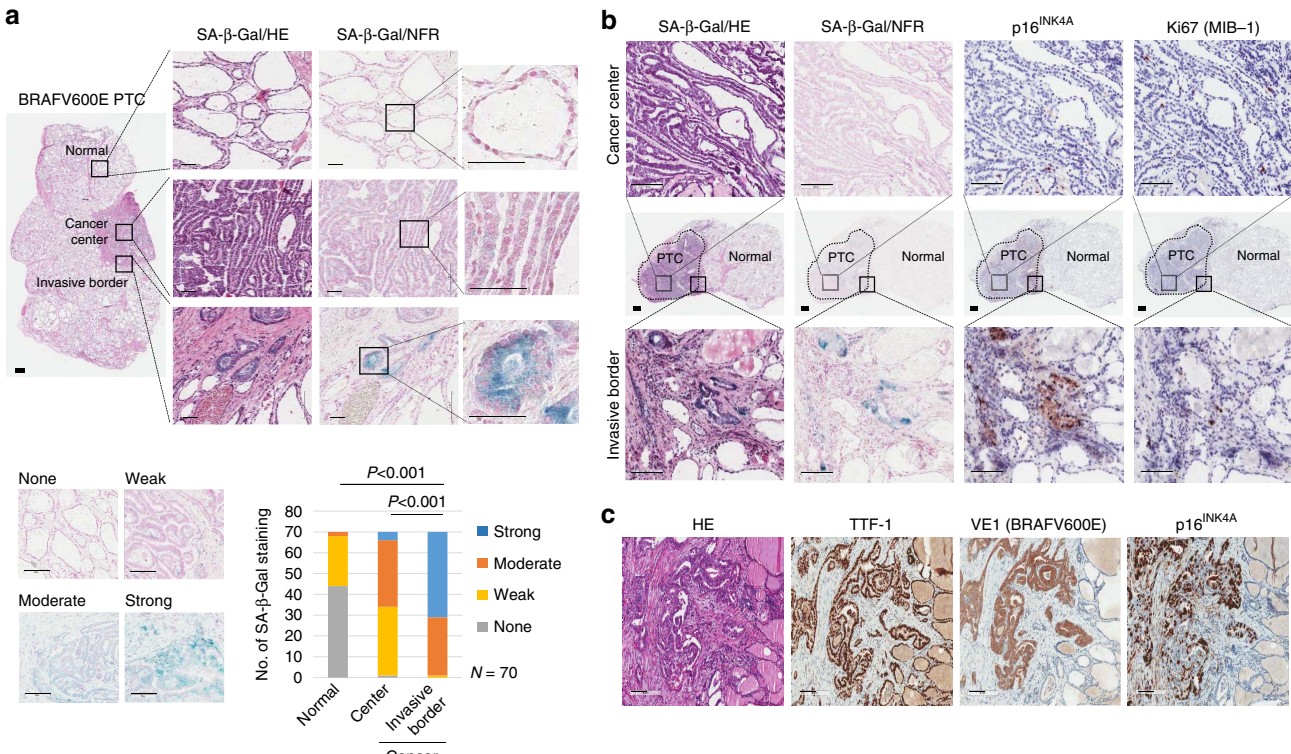

**Figure 1 | Senescent tumour cells are located at the front region of invasion in BRAFV600E-expressing PTC.** (**a**) PTC frozen sections were stained with SA-β-Gal, HE or nuclear fast red as counterstains, and images of a normal follicle, centre and invasive area of cancer were acquired (Upper panel, thick bars indicate 1 mm and thin bars indicate 50 μm). Seventy cases of BRAFV600E-expressing PTC were stained with SA-β-Gal and the intensity was analysed and presented as none, weak, moderate or strong. The P value shown was calculated by $\chi^2$ test (Lower panel, bars indicate 100 μm). (**b**) BRAFV600E-expressing PTC frozen tissues were sectioned serially and stained for SA-β-Gal, p16$^{INK4A}$ and Ki67 (MIB-1). Upper panel shows the cancer centre area, and lower panel shows the cancer invasive area. (**c**) The senescent program is present in neoplastic epithelial cells of BRAFV600E-expressing PTC. p16$^{INK4A}$, BRAFV600E (VE1) and TTF-1 expression was analysed in paraffin embedded tissues. Thick bars indicate 1 mm and thin bars indicate 100 μm in **b**,**c**.

cells undergo senescence after migration/invasion is present, these data suggested that senescent cells exhibited a higher invasive ability than non-senescent tumour cells. Senescent cells acquire many changes in gene expression, mostly as alterations in mRNA expression, resulting in increased expression of numerous proteins, a phenomenon referred to as SASP. Within the SASP, we focused on the expression of proteases and chemokines, since senescent tumour cells face with ECM at the invasive border. We first evaluated SASP expression in the entire cancer area of PTC tissues which were composed of senescent and non-senescent tumour cells and compared the results with those of corresponding non-neoplastic thyroid tissues by real-time PCR. Although wide variations in SASP expression existed across PTC cases, the expression of interleukin (IL)-6, IL-8 and matrix metalloproteinases (MMPs) was markedly upregulated in cancer tissue as compared with levels observed in corresponding non-neoplastic thyroid tissues (Fig. 3b, Supplementary Table 1). Some PTC cases exhibiting high SASP expression shows more frequent lymph metastases, but there were no differences in histologic grade between PTC cases. To investigate SASP expression in senescent cells more precisely, we compared the genes expression profiles between the cancer centre and invasive borders ( × 10 microscopically dissected from the cancer margin by a pathologist; n = 3), where senescent cells were predominantly present, by RNA sequencing and detected an increase in SASP expression in the cancer invasive area (Fig. 3c, right panel). We validated SASP expression using real-time PCR analysis (Fig. 3c, left panel). Furthermore, we performed immunohistochemical staining of MMPs in ten cases

of PTC, and found that immunoexpression of MMPs was higher than levels observed at the cancer centre, which agreed with mRNA expression (Fig. 3d). Taken together, these data indicated that senescent tumour cells in BRAFV600E-expressing PTC exhibited invasive ability and expressed SASP, including MMPs. Since we could not separate the senescent tumour cells from the primary cancer, we employed an *in vitro* senescent model to mimic the senescence phenotype occurring in BRAFV600E-expressing PTC. We generated a *BRAFV600E*-mutant lentivirus and induced OIS in isolated primary human thyrocytes, and further confirmed that the senescence phenotype was diminished by *shBRAF* lentivirus infection (Fig. 3e). The gene expression profiling in *BRAFV600E*-induced senescent thyrocytes showed similar result as those of senescent tumour cells in PTC (Fig. 3f,g and Supplementary Fig. 4). In addition, we found that *BRAFV600E*-induced senescent thyrocytes exhibited a higher invasive ability (Supplementary Fig. 5). Therefore, we considered that our *in vitro* senescent model might be useful for determining of the roles of senescent cells in BRAFV600E-expressing PTC.

**Senescent cells in lymphovascular invasion and metastasis.** During collective cell migration and invasion, a coordinated movement between leader and follower cells is essential, and the guidance chemical or mechanical signals are required for directional migration[18]. First, we attempted to evaluate the mechanical guidance signals by analysing polarization of the actin cytoskeleton in PTC, but failed to identify this polarization (Supplementary Fig. 6). Since senescent cells produce abundant

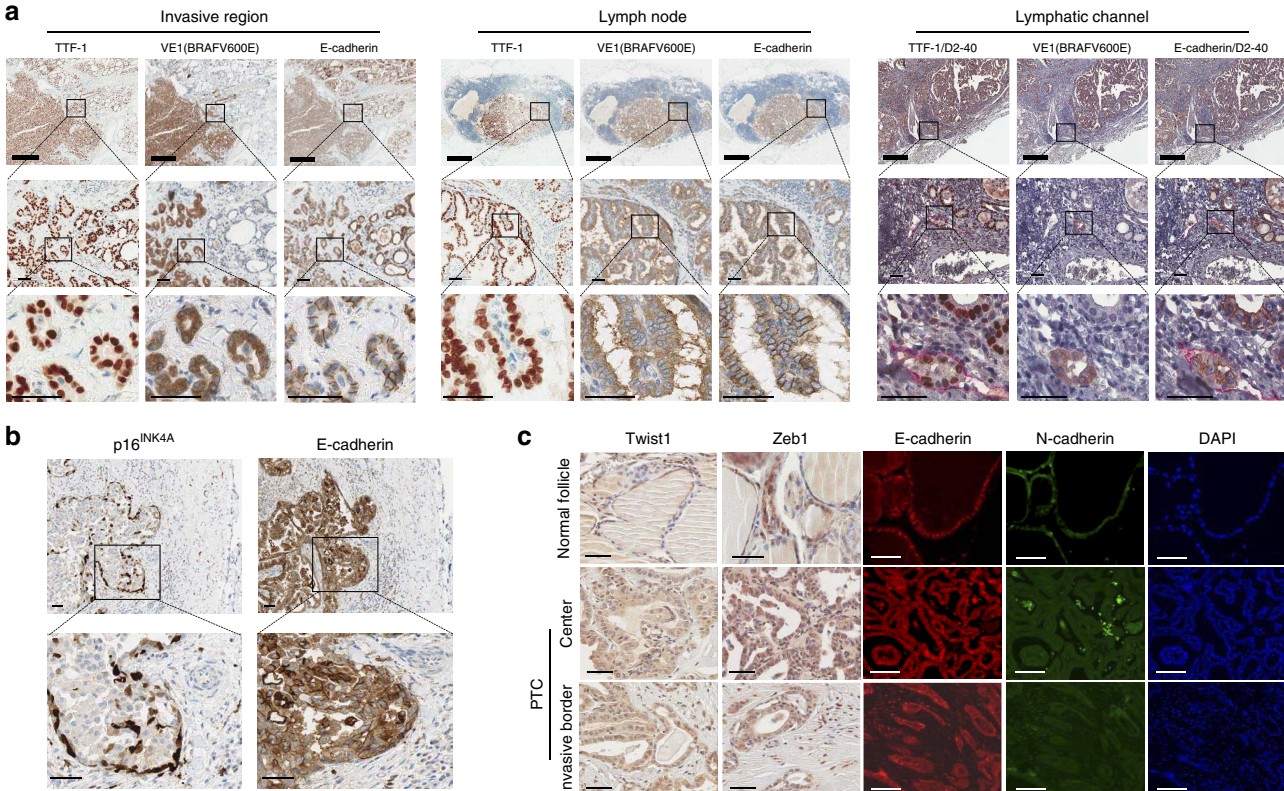

**Figure 2 | Senescent tumour cells in the collective invasion.** (**a**) Collective invasion in BRAFV600E expressing PTC. Invasive region of BRAFV600E-expressing PTC with metastatic lymph nodes and tumour emboli in the lymphatic channel were immunostained with TTF-1, BRAFV600E, E-cadherin, D2-40 (lymphatic vessel marker, red color). (**b**) p16INK4A-immunopositive senescent tumor cells located at the front region of E-cadherin conserved collective invasion. (**c**) Twist1, Zeb1, E-cadherin, and N-cadherin expression was analyzed in paraffin embedded tissue sections from 20 cases of BRAFV600E-expressing PTC, with representative results are presented. Thick bars indicate 1 mm (**a**) and thin bars indicate 50 μm (**a**,**b**,**c**), respectively.

SASP, including chemokines, we hypothesized that the chemical guidance signals are involved in the collective cell migration and invasion of PTC (Fig. 4a). As shown in Fig. 4b, the cluster of senescent tumour cells was located in the front region of collective invasion, whereas non-senescent tumour cells were located in the rear area of the senescent cells, appearing like that senescent tumour cells led the invasion, followed by non-senescent cells. We then investigated whether senescent cells were present during metastasis: senescent cells were actively involved in the cancer invasion and metastasis. Therefore, we performed SA-β-Gal staining, p16INK4A and VE1 immunohisto-chemistry in 10 cases of PTC with metastatic lymph nodes and found that p16INK4A immunopositive senescent cells existed in the metastatic foci of lymph nodes in all PTC cases (Fig. 4c, Supplementary Table 2). We then identified SA-β-Gal-positive senescent cells in tumour emboli of peritumoral lymphovascular spaces in primary PTCs. p16INK4A and VE1 immunohisto-chemistry indicated that BRAFV600E-expressing tumour emboli in lymphatic channels included p16INK4A-immunopositive senescent cells (Fig. 4d). These data demonstrated that BRAFV600E-expressing senescent cells were not temporarily present during the initial process of collective invasion, but existed during the collective lymphovascular invasion and metastasis.

**CXCL12/CXCR4 signalling in the collective invasion.** We then determined whether senescent tumour cells in PTC could lead the collective invasion. To address this question, we performed an organoid three-dimensional (3D) culture assay using primary tumour tissues from BRAFV600E-expressing PTC as previously described[6]. After the emergence of collectively migrating cells from the tumour organoids, we analysed the tumour organoid by SA-β-Gal staining. SA-β-Gal positive senescent cells were identified at the front of invasive foci (Fig. 5a), with 77% of leading cells showing SA-β-Gal positivity in organoid cultures. To verify that the senescent cells induced chemogradient collective invasion (Fig. 4a), we analysed chemokines expression at the invasion border of PTC *in vivo* and *BRAFV600E*-induced senescence *in vitro* model by RNA sequencing. RNA-sequencing analysis revealed upregulation of several chemokines, among which we focused on the expression of *CXCL* family, since C-X-C-motif ligand (CXCL)/CXCR signalling is an essential mechanism of collective invasion and lymph node metastasis[19]. We found that most of *CXCLs/CXCRs* and chemokine ligands (CCLs)/chemokine receptors (CCRs) showed no expression or very lower FPKM (Fragments Per Kilobase of transcript per Million fragments mapped) values in PTC (Supplementary Table 3). We further analysed RNA expression ratios between the cancer centre and the invasive border and found increased expression of several chemokines in the invasive area (Fig. 5b). Although several chemokines, such as *CCL4*, *CCL19*, *CCL21*, *CXCL2* or *CXCL12*, were upregulated at the invasive border than cancer centre, expressions of their receptors were very low in PTC, except for *CXCL12* receptor (Supplementary Tables 3 and 4). We further investigated the expression of several members of the *CXCLs/CXCRs* family in 13 cases of BRAFV600E-expressing PTC tissues by real-time PCR and found that *CXCL1*, *CXCL2* and *CXCL12* expression was

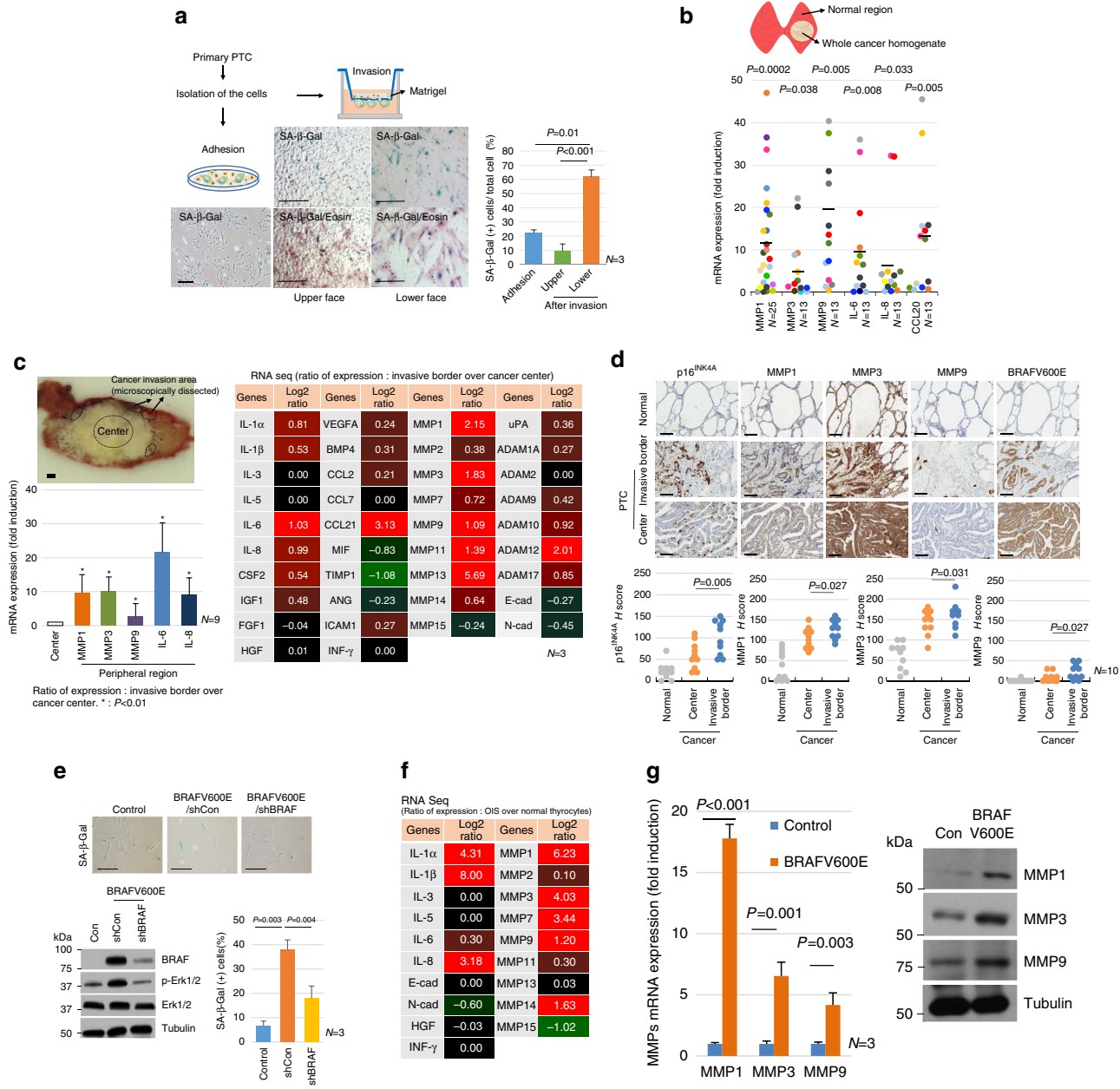

**Figure 3 | Senescent cells exhibit high invasive ability via SASP expression.** (**a**) Tumour cells were isolated from BRAFV600E-expressing PTC, and the invasion ability of senescent cells was then analysed. Tumour cells were seeded in 60-mm dishes or transwells and maintained for 24 h. Cells were stained with SA-β-Gal in 60-mm dishes or SA-β-Gal/eosin in transwell membranes, followed by analysis of percentage of senescent cells (n = 3). (**b**) SASP expression in BRAFV600E-expressing PTC. SASP expression was analysed in BRAFV600E-expressing PTC and adjacent normal tissues by real-time PCR and presented as a dot graph (n = 13). Values indicate the relative value compared to that of the normal follicle. In the case of *MMP1*, 25 cases were analysed. (**c**) RNA-sequencing analysis in the centre and invasive area of PTC. The central or invasive region of cancer tissues were obtained after surgical resection by a pathologist followed by analysis of target mRNA expression by RNA sequencing (n = 3, average value, right panel) or real-time PCR (n = 9, left lower panel). The values in RNA sequencing and real-time PCR indicate the Log2 ratio and the relative values of cancer invasive region compared with the centre of the cancer, respectively. (**d**) Immunohistochemical analysis of MMP1, MMP3, MMP9 and p16INK4A expression in BRAFV600E-expressing PTC (n = 10), and analysed using H score. (**e**) *BRAFV600E*-induced senescence in primary normal thyrocytes. Normal thyrocytes were infected with control, *BRAFV600E*/shCon or *BRAFV600E/shBRAF* lentivirus for 10 days. SA-β-Gal staining (n = 3), BRAF and p-Erk1/2 western blots were performed. (**f**) RNA-sequencing analysis in normal and *BRAFV600E*-induced senescent thyrocytes. Total RNA was isolated from normal or *BRAFV600E*-induced senescent thyrocytes, and RNA sequencing was performed. The value indicates the Log2 ratio compared with RNA expression of normal thyrocytes (n = 2, average value). (**g**) *MMPs* expression was analysed in normal and *BRAFV600E*-induced senescent thyrocytes by real-time PCR (n = 3, left panel) and western blotting (right panel). The P value shown (**d**) was calculated by Wilcoxon signed rank test and the others were calculated by Student's *t*-test. Error bars, s.d. Thick bars indicate 1 mm (**c**), and thin bars indicate 100 μm.

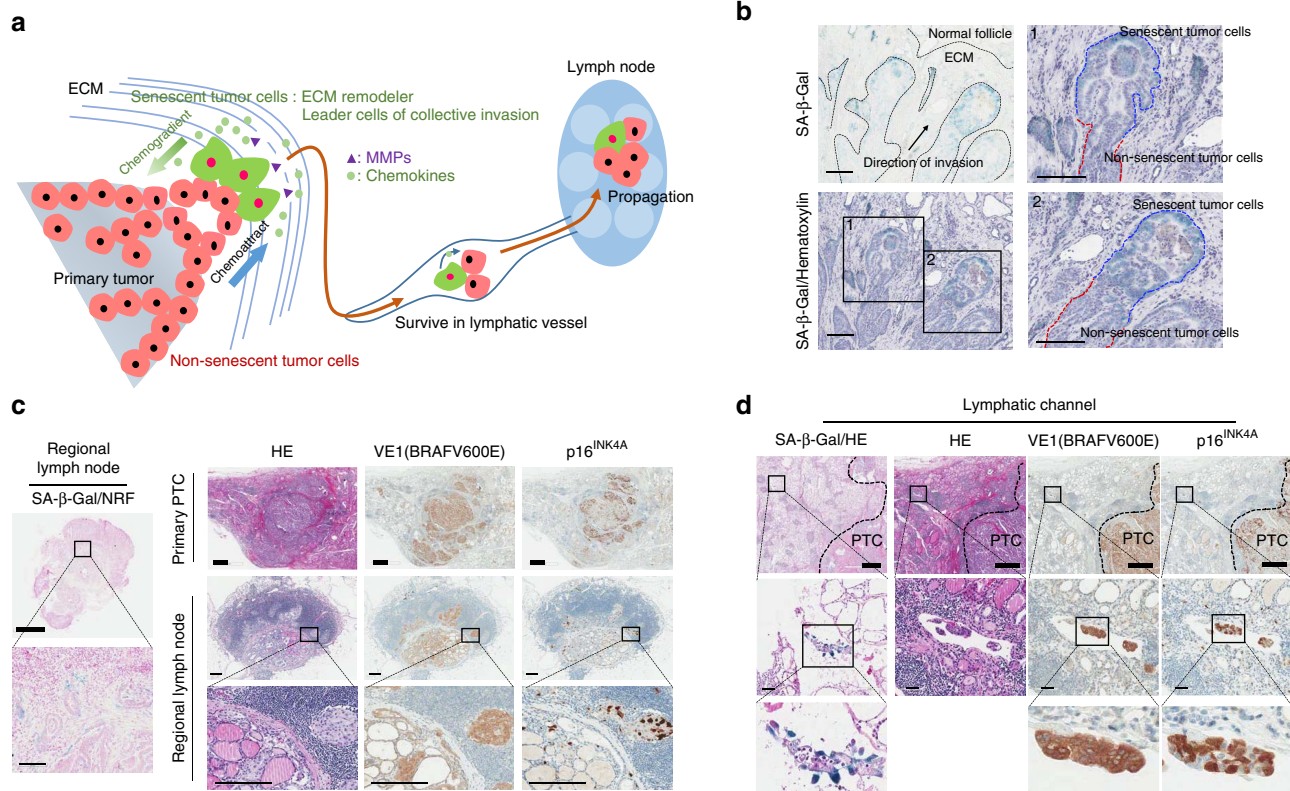

**Figure 4 | Senescent tumour cells involved in cancer invasion and lymph node metastasis. (a)** Schematic representation of the role of senescent tumour cells in collective invasion and lymph node metastasis. **(b)** Senescent tumour cells located in the front region of collective invasion. PTC frozen sections were stained with SA-β-Gal only (left upper) and SA-β-Gal/hematoxylin (left lower). The black dotted line indicates the margin of cancer (left upper panel), and the blue and red dotted lines indicate senescent and non-senescent tumour cells (right panel), respectively. '1' and '2' indicate the high-magnification fields of the original figure. **(c)** Senescent tumour cells in regional lymph nodes. Ten cases of primary PTC and their regional lymph nodes were analysed by SA-β-Gal (left panel), p16INK4A and BRAFV600E (right panel). **(d)** Senescent tumour cells in lymphatic channels. SA-β-Gal, BRAFV600E and p16INK4A expression in tumour emboli was analysed in lymphovascular vessels. Thick bars indicate 1 mm (**c,d**) and thin bars indicate 50 μm (**d**), 100 μm (**b,c**) left panel and 200 μm (**c**) right panel, respectively.

significantly upregulated in the cancer area compared with the normal tissue ($P = 0.02$, $P = 0.004$ and $P = 0.03$, respectively, Student's $t$-test), whereas, among the receptors of these chemokines, only *CXCR4* was significantly upregulated ($P = 0.02$, Student's $t$-test; Fig. 5c, left panel). We further analysed mRNA expression of *CXCL12* and *CXCR4* at the invasive border and cancer centre, finding that *CXCL12* expression was significantly increased at the invasive border ($P < 0.01$, Student's $t$-test; Fig. 5c, right panel), indicating that CXCL12/CXCR4 signalling was significantly upregulated at the invasive border and resulting in gradients of expression between the cancer centre and invasive border.

We performed p16INK4A and CXCL12/CXCR4 immunohistochemistry on tissue from 13 cases of BRAFV600E-expressing PTC to evaluate the relationship between senescent cells and CXCL12/CXCR4 expression. The expression of CXCL12, the unique CXCR4 ligand, significantly increased in the collective invasion area where p16INK4A-immunopositive senescent cells were present, whereas CXCR4 was diffusely overexpressed in all cases of cancer tissues (Fig. 5d). These data revealed that senescent cells might induce a CXCL12-mediated chemoattractant gradient between leader (senescent tumour cells) and follower (non-senescent tumour cells) cells. An *in vitro* BRAFV600E-induced senescence model showed marked induction of *CXCL12* mRNA expression and secretion of CXCL12 protein in medium (Fig. 5e).

**Senescent cells lead the collective invasion**. To determine whether senescent cells were capable of attracting cancer cells through the CXCL12/CXCR4 signalling, we performed an *in vitro* cell migration assay using two kinds of thyroid cancer cell lines: SNU790-*CXCR4* and HTH83 (Supplementary Fig. 7). *BRAFV600E*-induced senescent thyrocytes enhanced the migratory ability of CXCR4-expressing SNU790 and HTH83 cells (Fig. 6a, Supplementary Fig. 8), and results of a transwell assay were consistent with those of the cell migration assay (Fig. 6b, Supplementary Figs 9 and 10). To rule out the possibility that other cytokines were involved in cancer cell migration, we created two kinds of *CXCL12* knockdown senescent cells by shRNA or application of the AMD3100 which is well known CXCR4 antagonist (Supplementary Fig. 11). Our results indicated marked inhibition of cancer cells attraction by inhibition of CXCR4 signalling (Fig. 6a,b, Supplementary Figs 8 and 10). These data demonstrated that *BRAFV600E*-induced senescent thyrocytes can attract thyroid cancer cells through CXCL12/CXCR4 signalling.

We next determined whether CXCL12 expressing senescent cells could serve as leaders of collective invasion. To address this question, we first performed a 3D-matrix invasion assay (Fig. 6c, upper panel). When SNU790-*CXCR4* thyroid cancer cells were co-cultured with normal thyrocytes, cancer cells invaded slowly with broad margins. However, in the presence of *BRAFV600E*-induced senescent thyrocytes, we observed protruding multicellular invasive strands from the tumour margin,

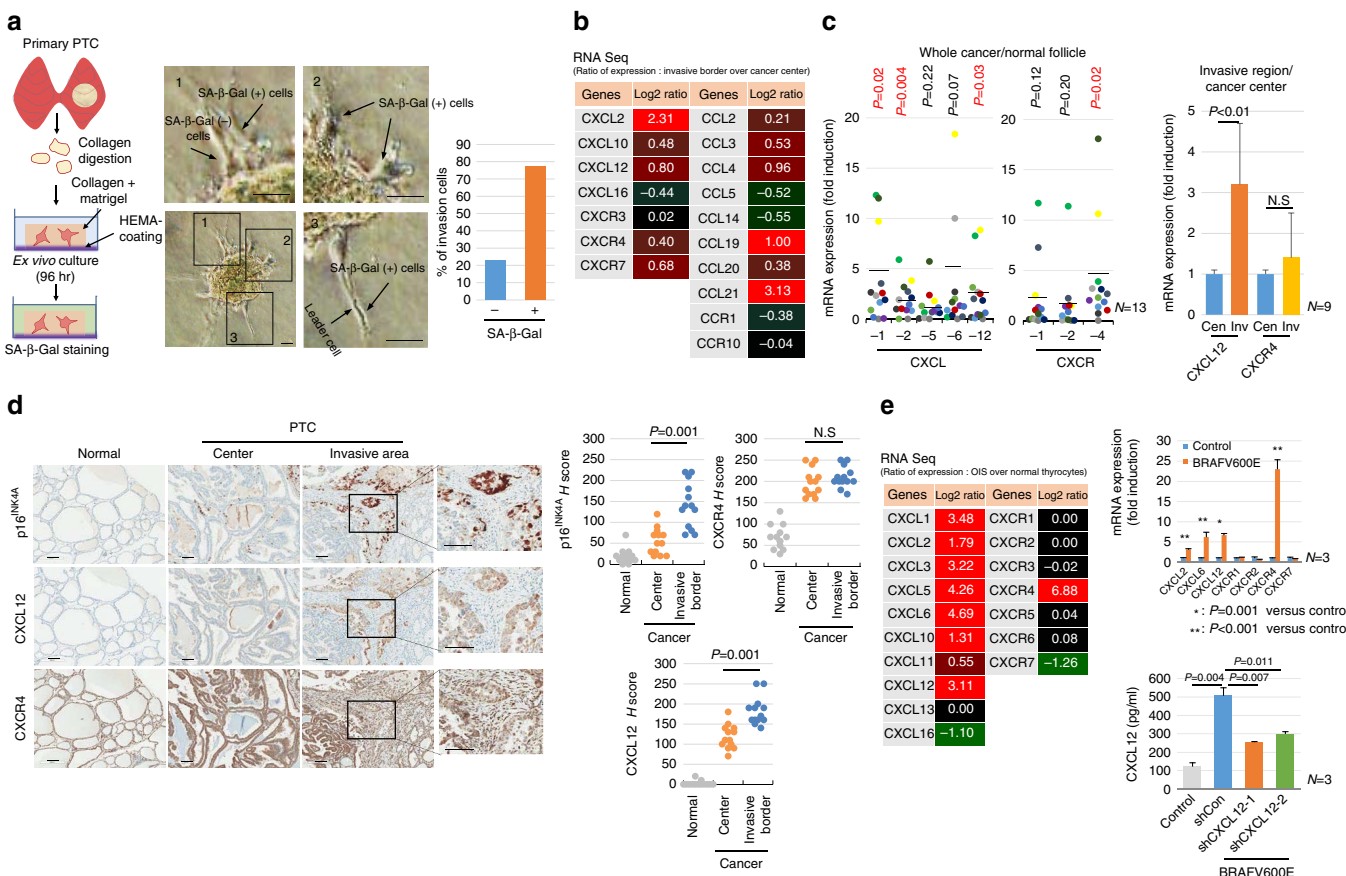

**Figure 5 | CXCL12/CXCR4 signalling in the collective invasion.** (**a**) Scheme of a leader-cell assay. PTC tissues were digested with collagenase to obtain small fragments of cancer tissues, and fragments were cultured in collagen I containing matrigel for 96 h, followed by SA-β-Gal staining (left panel). SA-β-Gal-positive migrating cells emerged from the tumour organoid (right panel). '1', '2' and '3' indicate the high-magnification field of the original figure. (**b**) CXCLs/CCLs and their receptor expression in cancer invasive region. Experimental scheme was same as Fig. 3c. Raw data of mRNA expression is summarized in Supplementary Table 3. (**c**) Expression of CXCLs and their receptors in BRAFV600E-expressing PTC. Expression of CXCLs and CXCRs was analysed in the normal region and PTC by real-time PCR and represented as a dot graph ($n = 13$, left panel). The values indicate the relative value compared to that of a normal follicle. Expression of CXCL12 and CXCR4 was analysed in the centre and invasive area of cancer by real-time PCR and represented as a bar graph ($n = 9$, right panel). 'Cen' and 'Inv' indicate the centre and invasive area of cancer, respectively. (**d**) Immunohistochemical analysis of CXCL12, CXCR4 and p16[INK4A] expression in BRAFV600E-expressing PTC ($n = 13$). Normal, centre and collective invasive regions of cancer were serially sectioned, and CXCL12, CXCR4 and p16[INK4A] expression was analysed by H score. 'N.S' indicates not significant. (**e**) Expression of CXCLs/CXCRs in BRAFV600E-induced senescent thyrocytes ($n = 2$, average value). Experimental scheme was same as Fig. 3f. Secreted CXCL12 protein was measured by ELISA ($n = 3$, right lower panel). The P value shown (**d**) was calculated by Wilcoxon signed rank test and the others were calculated by Student's t-test. Bars indicate 50 μm (**a**), 100 μm (**d**), respectively. Error bars, s.d.

indicating collective invasion (Fig. 6c, lower panel). We then performed live image analysis of the *BRAFV600E*-induced senescent thyrocytes/thyroid cancer cell co-culture system. When normal primary thyrocytes (green fluorescent protein (GFP)) and thyroid cancer cells (mCherry) were co-cultured, only a small number of thyroid cancer cells (mCherry) invaded the matrigel (Fig. 6d, upper panel), however, in co-cultured *BRAFV600E*-induced senescent thyrocytes (GFP) and cancer cells (mCherry), the *BRAFV600E*-induced senescent thyrocytes invaded the matrigel as leader cells, with formation of a canal (bright field), and cancer cells (mCherry) followed senescent cells along the canal (Fig. 6d, middle panel, high magnification field, Supplementary Fig. 12). When *CXCL12* in senescent cells was knocked down by shRNA, the invasion of cancer cells was markedly decreased (Fig. 6d, lower panel, Supplementary Fig. 12). These data demonstrated that *BRAFV600E*-induced senescent thyrocytes lead collective invasion through CXCL12/CXCR4 signalling.

**CXCL12/CXCR4 signalling is involved in anoikis resistance.** We further analysed CXCL12/CXCR4 signalling in cancer progression. In lymphovascular spaces, cancer cells are detached from the ECM, and a loss of cell-matrix interactions induces apoptosis, referred to as anoikis. Therefore, the acquisition of anoikis resistance is a critical step required for metastasis, and anoikis-resistant tumour cells are associated with high incidence of metastatic lesions and increased cell survival in the blood[20]. EMT is among the intrinsic mechanisms used to escape anoikis, however, in this study, we observed well-preserved E-cadherin expression in tumour emboli, implying that EMT is not the principal mechanism in PTC (Fig. 7a). Interestingly, p16[INK4A]-positive senescent cells in CXCR4-expressing tumour cells expressed CXCL12 (Fig. 7a,b, Supplementary Fig. 13). CXCR4 and its ligand, CXCL12, can promote metastasis by preventing anoikis in cancer cells[21]. To determine whether senescent cells could increase the anoikis resistance of thyroid cancer cells, we performed an anoikis resistance assay using HEMA-coated plates (Fig. 7c, upper panel).

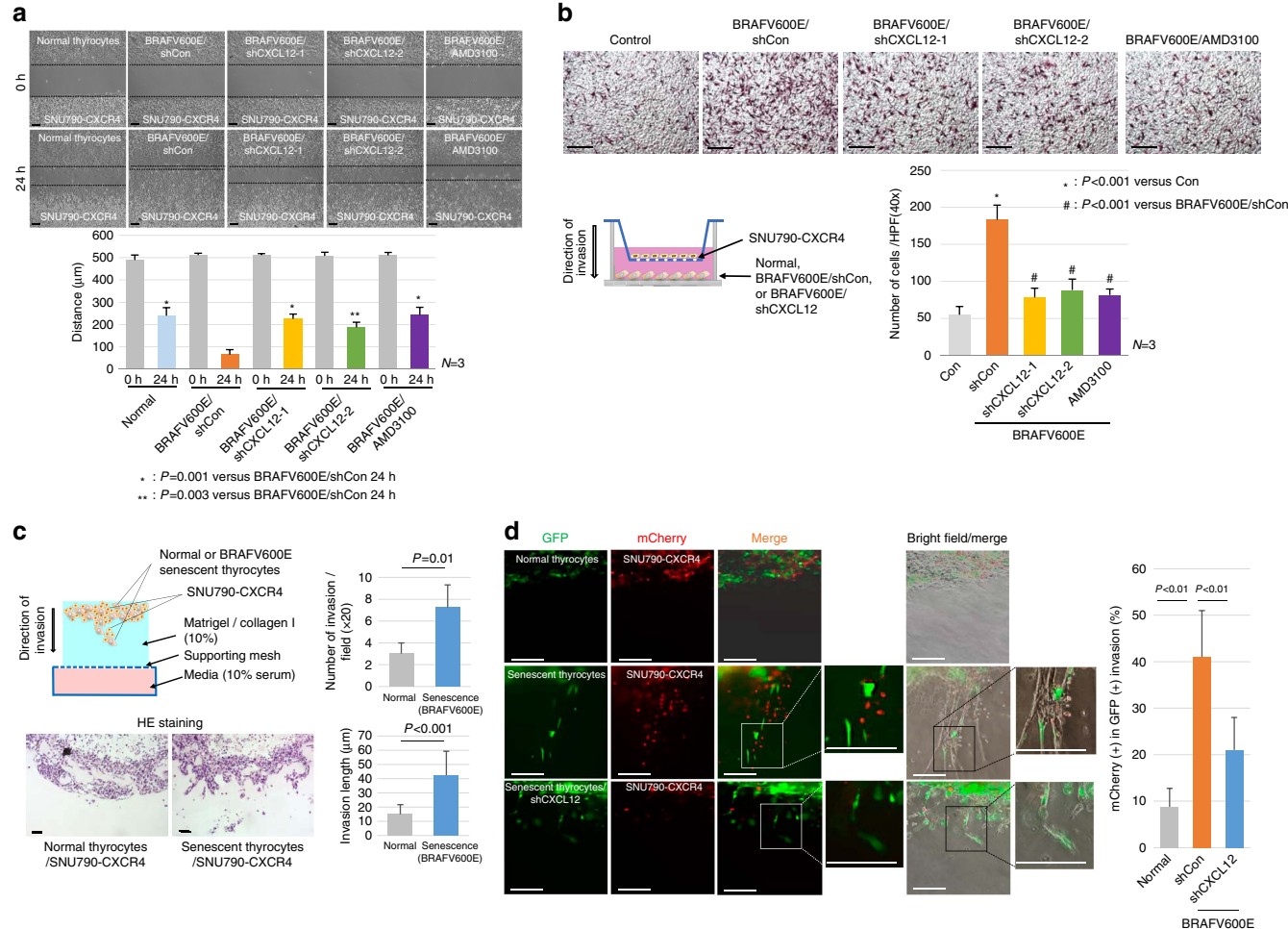

**Figure 6 | Senescent cells lead collective invasion.** (**a**) *In vitro* cell-migration assay. Normal/SNU790-*CXCR4*, *BRAFV600E*/SNU790-*CXCR4* and *BRAFV600E-shCXCL12*/SNU790-*CXCR4* cells were seeded. After 24 h, cell migration was measured. One set of *BRAFV600E*/SNU790-*CXCR4* cells was treated with 1μM of AMD3100. Bar graph indicates the average of independent measurements (*n* = 3). (**b**) Transwell assay. SNU790-*CXCR4* cells suspended in medium were seeded in transwell. Control, *BRAFV600E*, *BRAFV600E*/*shCXCL12* or *BRAFV600E*/AMD3100 treated cells were seeded at the bottom. After 24 h, cells that invaded the lower surface of the filters were counted. The number of migrated cells was counted in the 40-fold magnification field, and presented in the bar graph (*n* = 3, right panel). (**c**) Three-dimensional invasion assay. SNU790-*CXCR4* cells were co-cultured with normal or *BRAFV600E*-induced thyrocytes on the top of collagen I containing matrigel for 48 h (upper panel). Cell invasion was assessed by HE staining (lower panel). Bar graph indicates the average of independent experiments (*n* = 3). (**d**) mCherry lentivirus-infected SNU790-*CXCR4* cells were co-cultured with GFP lentivirus-infected normal (upper panel), *BRAFV600E* (middle panel) or *BRAFV600E*/*shCXCL12* thyrocytes (lower panel) on the top of collagen I containing matrigel for 48 h. Independent experiments were performed and data are presented in the bar graph (*n* = 3). The *P* values were calculated by Student's *t*-test. Bars indicate 50 μm (**c**,**d**) and 100 μm (**a**,**b**), respectively. Error bars, s.d.

When we cultured thyroid cancer cells alone (SNU790-*CXCR4* or HTH83), anoikis was rapidly induced in cancer cells in suspension. However, when we co-cultured thyroid cancer cells with senescent thyrocytes, the survival of cancer cells increased after a prolonged period of forced detachment. Furthermore, *CXCL12* knockdown by application of two different *shCXCL12* or the CXCR4 antagonist AMD3100 decreased the survival of cancer cells (Fig. 7c, lower panel, Supplementary Fig. 14a). We further analysed anoikis resistance by assessing caspase and apoptosis-related proteins expression, with results confirming that *BRAFV600E*-induced senescent thyrocytes increased the anoikis resistance of thyroid cancer cells (Fig. 7d,e and Supplementary Fig. 14b,c). These data revealed that senescent cells in tumour emboli of PTC can increase anoikis resistance through CXCL12/CXCR4 signalling.

**Senescent cells enhance collective lymphovascular invasion.** To verify the invasion and metastatic ability of senescent cells *in vivo*,

thyroid cancer cells together with *BRAFV600E*-induced senescent or normal thyrocytes were orthotopically transplanted into the thyroid gland of female nude mice (Fig. 8a). Since the PTC cell line (SNU790, SNU790-*CXCR4*) did not form a tumour mass in nude mice (Supplementary Fig. 15), we used the anaplastic thyroid cancer cell line (HTH83; *BRAF* wild type and *HRASQ61R*$^{+/-}$)[22], which exhibited higher *CXCR4* expression (Supplementary Fig. 7). We observed that 58% (14/24) of nude mice developed thyroid cancer (6/10, normal thyrocytes/cancer transplanted group; 8/14, *BRAFV600E*-induced senescent thyrocytes/cancer transplanted group; Fig. 8b). There was no significant difference in terms of tumour development (60% versus 57%) and tumour size (7.5 ± 2.81 mm versus 6.7 ± 3.64 mm; *P* = 0.414, Mann–Whitney *U*-test) between groups (Fig. 8c). We also examined neck lymph nodes and lymphovascular invasion in the peritumoral region of transplanted mice. Metastatic lymph nodes were detected in both groups (cancer positive lymph node/total identified lymph node; control, 1/14 (7%); senescence, 3/26 (12%), but no statistical

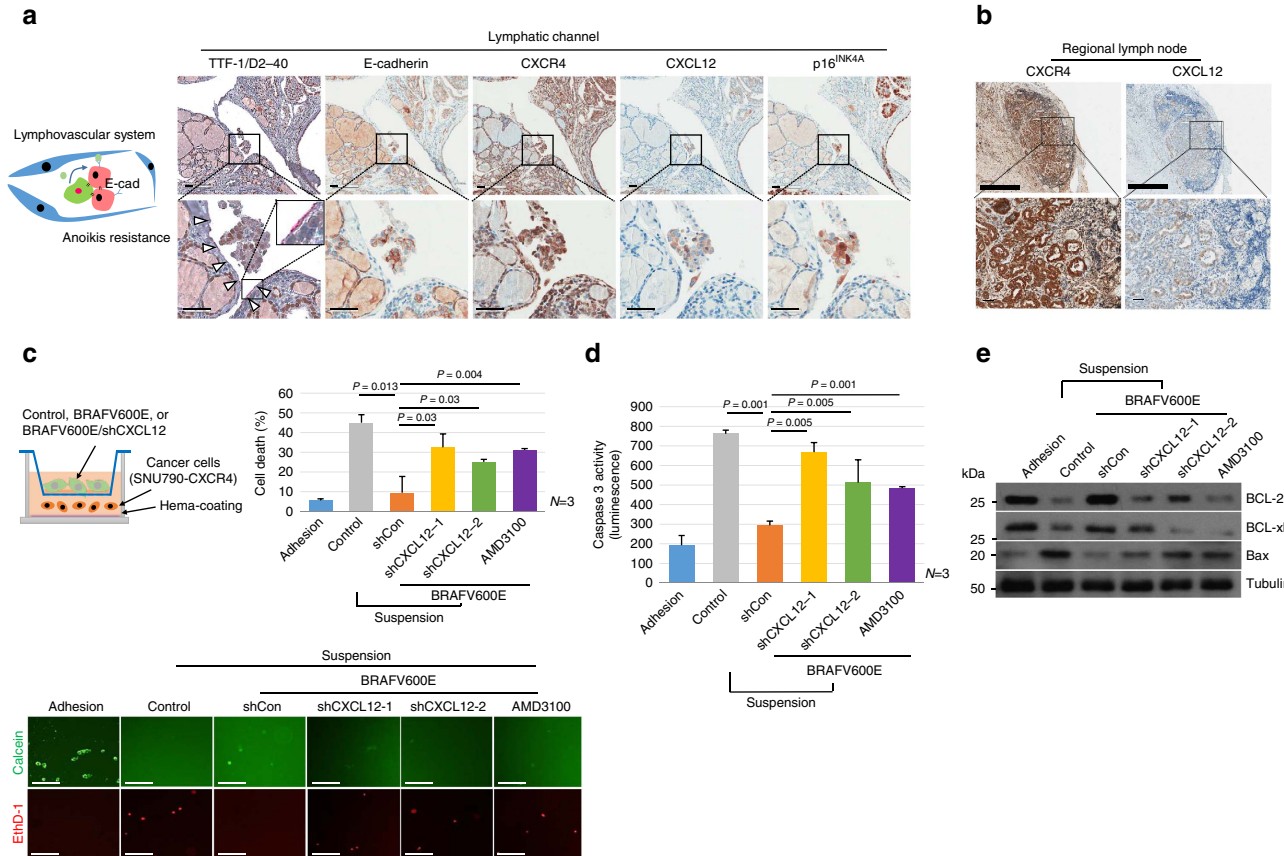

**Figure 7 | Senescent cells confer anoikis resistance.** (**a**) The epithelial marker E-cadherin is retained in cancer emboli in lymphatic channels. PTC specimens were serially immunostained with TTF-1 (brown colour in nuclei)/D2-40 (red colour in cytoplasm), E-cadherin, CXCR4, CXCL12 and p16$^{INK4A}$. White triangles indicated D2-40 stained lymphatic vessels. (**b**) Metastatic tumour cells at lymph nodes were stained with CXCR4 and CXCL12, respectively. (**c**) Anoikis inhibitory function of senescent cells. Control, *BRAFV600E*/shCon, *BRAFV600E/shCXCL12* or *BRAFV600E*/AMD3100 treated cells were co-cultured with thyroid carcinoma cells (SNU790-*CXCR4*) in HEMA-coated plates for 12 h, and cell death was determined by Calcein AM and EthD-1 staining, (**d**) caspase activity and (**e**) apoptosis related proteins expression. Independent experiments were performed and data are presented in the bar graph (n = 3). Thick bars indicate 1 mm (**b**) and thin bars indicate 50 μm (**a–c**), respectively. The *P* values were calculated by Student's *t*-test. Error bars, s.d.

significance was observed between the groups. Interestingly, lymphovascular invasion by tumour cells was more frequent in *BRAFV600E*-induced senescent thyrocytes/cancer transplanted mice than in normal thyrocytes/cancer transplanted mice, as we identified 22 foci of lymphovascular invasion in *BRAFV600E*-induced senescent thyrocytes plus cancer transplanted mice, but only 6 foci of lymphovascular invasion in normal thyrocytes plus cancer transplanted mice (P = 0.043, Mann–Whitney *U*-test) (Fig. 8d). In addition, BRAFV600E-expressing senescent thyrocytes were observed in a multicellular unit of tumour emboli in lymphovascular channels (Fig. 8e). Collectively, these data suggested that BRAFV600E-expressing senescent thyrocytes might be associated with increased lymphovascular invasion and increased anoikis resistance in thyroid cancer.

## Discussion

Cellular senescence can be induced by loss of telomeres after extensive proliferation, as well as exposure to a variety of stresses, such as oxidative stress, DNA-damaging agents or oncogene activation[9,23–27]. Senescence, especially OIS, is considered as a barrier to tumorigenesis, given that OIS is commonly observed in premalignant tumours, such as melanocytic nevi, lung adenomas or pancreatic intraductal neoplasia, but is rare in their malignant counterparts[9,28,29]. However, previous studies

indicated that a significant proportion of senescent cells are present in various types of cancers, such as B cell lymphoma, lung, breast, colorectal and thyroid cancers[11–16,28,30–32], indicating that senescent cells do not completely disappear. We also found that senescent cells were present in BRAFV600E-expressing PTC. Cellular senescence in cancer cells can be induced by cell-autonomous mechanisms, such as overexpression of oncogenes or loss of tumour-suppressor genes[33–35]. In addition, certain types of tumour-infiltrating immune cells can induce senescence in a non-cell-autonomous manner[36–38]. To investigate the role of cellular senescence in PTC, we utilized a *BRAFV600E*-induced senescence *in vitro* model, a cell-autonomous senescence model, for several reasons. First, *BRAFV600E* mutation is the most common driving oncogene mutation in PTC tumorigenesis. Second, all senescent cells in BRAFV600E-expressing PTCs expressed BRAFV600E protein. Last, senescent cells could be identified in the areas of rare or no infiltration of immune cells.

Senescent cells present a steady status, typically arrested in the G1 phase of the cell cycle, but metabolically and transcriptionally active, secreting a group of factors collectively termed the SASP[10]. The composition and amounts of the SASP may vary according to the duration of senescence and mechanisms inducing senescence[10,39]. In the present study, certain types of SASP, such as MMPs and chemokines, identified in our *in vitro* model

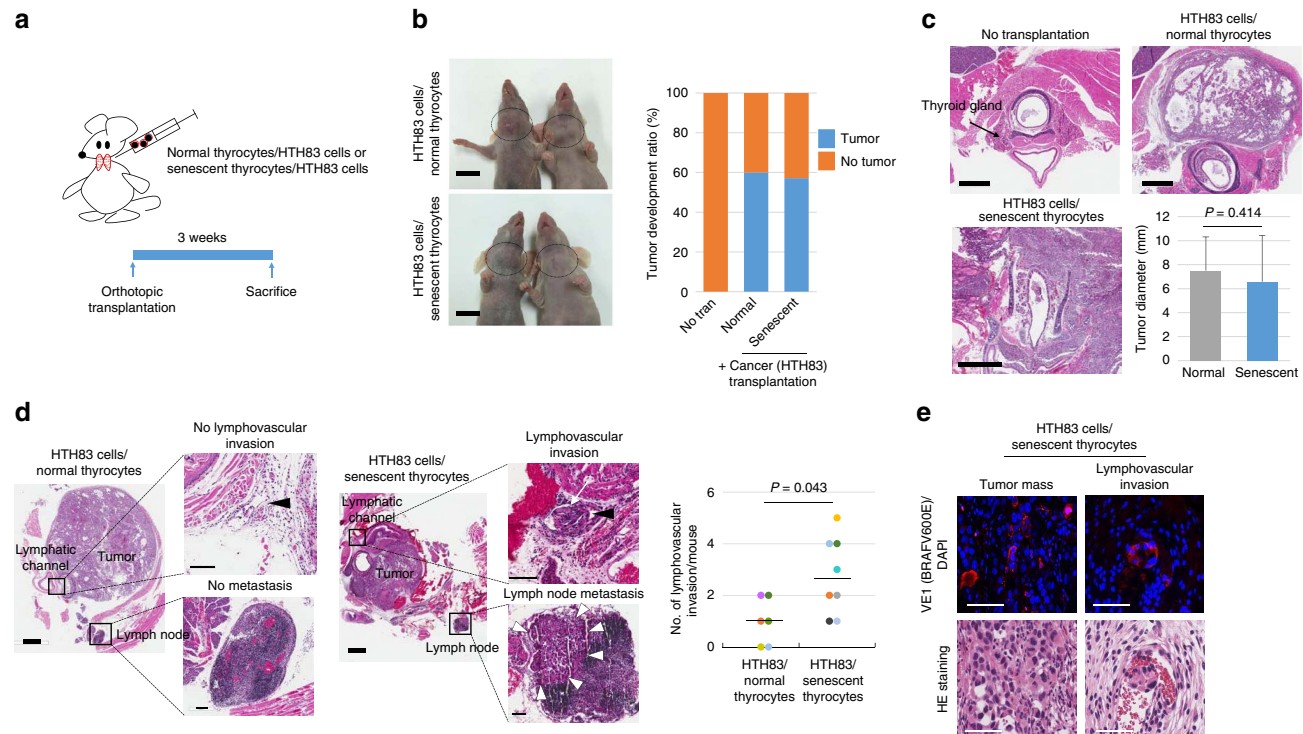

**Figure 8 | Senescent cells involved in collective lymphovascular invasion in *in vivo* nude mice.** (**a**) Schematic representation of orthotopic transplantation. (**b**) Thyroid tumours developed in female nude mice. $1 \times 10^6$ cells ($9 \times 10^5$ HTH83 cells $+1 \times 10^5$ normal or *BRAFV600E*-induced senescent thyrocytes) were transplanted into the thyroid gland of 9-week-old female nude mice. The mice were euthanized after 3 weeks and the incidence of (**b**) tumour development, as well as (**c**) the tumour size, were analysed. The *P* values were calculated by Mann–Whitney *U*-test. Circles indicate the tumour mass in the thyroid. (**d**) Lymphovascular invasion and regional lymph node metastasis in transplanted mice. Lymphovascular invasion and lymph node metastasis were analysed in nude mice developing cancer. Lymphovascular invasion is described as the number of lymphovascular invasion in cancer developing mice. Black arrowheads indicate a lymphatic channel. White arrows indicate tumour emboli in lymphatic channel. White arrowheads indicate a cancer-positive regional lymph node. (**e**) BRAFV600E-expressing senescent cells in the tumour mass and tumour emboli in a lymphovascular vessel. Statistical analyses were performed by Mann–Whitney *U*-test. Thick bars indicate 1 cm (**b**), 1 mm (**c**,**d**) and thin bars indicate 50 μm (**e**), 100 μm (**d**), respectively. Error bars, s.d.

also overlapped significantly with those of *in vivo* senescent tumour cells. These data suggested that SASP expressed at the invasive borders of PTC could at least in part originate from BRAFV600E-expressing senescent cells. Among the SASP, we focused on chemokines and observed that various chemokines were upregulated in *in vitro* senescent cells. We deeply considered several kinds of possible chemokines signalling in cancer invasion and metastasis using RNA-sequencing analysis. Although several chemokines were upregulated at the invasive border of PTC, the expression of their receptors were very low, except for the CXCL12 receptor. These chemokine expression patterns were also identified in *in vitro* model (RNA-sequencing data was deposited in Gene Expression Omnibus: GSE81144 and GSE83560). Finally, we found that the expression of *CXCL12/CXCR4* was significantly upregulated in both senescent cells *in vitro* and in PTC tissues, therefore, we focused on CXCL12/CXCR4 signalling. CXCR4 activation stimulates the directed migration of cancer cells and increases their invasiveness[21]. Migration and invasion of CXCR4-expressing tumour cells depend upon a gradient of the chemokine, CXCL12, the only known ligand for CXCR4 (ref. 40). Interestingly, CXCL12 expression was not homogeneous across the tumour areas of PTC, but prominently increased in areas of collective invasion in PTC. Moreover, CXCL12-expressing cells were clearly correlated with senescent cells in areas of collective invasion. These data suggested that senescent cells in PTC can generate a chemotactic gradient at invasive borders through increased

CXCL12 expression. Generation of a front-rear CXCL12 gradient can guide collective migration through the activation of CXCR4 signalling, and beyond guidance, chemotactic signalling can reinforce cell-cell adhesion and cell clustering[18]. Here, we demonstrated that CXCR4-expressing PTC cells were able to follow CXCL12-expressing senescent thyrocytes, and that CXCL12-expressing senescent thyrocytes effectively attracted CXCR4-expressing cancer cells in the migration assay. In addition, 3D invasion assay showed that some senescent thyrocytes formed the route in collagen containing matrigel by expressing MMPs and subsequently led cancer cells. Interestingly, when we administered the CXCR4 antagonist (AMD3100) to the organoid culture, we observed marked decreases in senescent cells invasion ability (Supplementary Fig. 16a). We also found that *MMPs* expression was markedly downregulated in AMD3100 treated cells (Supplementary Fig. 16b), which was consistent with the results of previous reports[41,42]. Collectively, these data suggested that senescent cells work as a 'storming party' in the collective invasion of cancer through SASP, including MMPs and CXCL12/CXCR4 signalling.

PTC is a well-differentiated malignancy, mostly presenting an indolent clinical behaviour and a good prognosis. However, PTCs are frequently associated with lymph nodes metastasis. Regional lymph node metastases are identified in 30 to 50% of PTC cases at the time of initial diagnosis[43,44]. In our series of BRAFV600E-expressing PTC, lymph node metastasis was detected in 48 of 79 cases (61%) (Supplementary Table 5). Although *BRAFV600E* mutation is one of

the prognostic markers related to lymph node metastasis in PTC[45,46], mechanisms related to frequent lymph node metastasis in BRAFV600E-expressing PTC has not yet been clearly explained. Here, we found neoplastic senescent cells in metastatic foci of regional lymph nodes. This finding raised a question of whether senescent cells in metastatic foci were from the primary tumour together with cancer cells in the manner of collective invasion, or cancer cells metastasized from the primary lesion and then became senescent cells after arrival at the metastatic site in lymph nodes. To answer this question, we examined lymphatic channels, the middle part of metastasis, in PTC tissues and micrometastatic foci in the lymph nodes. SA-β-Gal and p16[INK4A] immunopositive neoplastic senescent cells were detected in tumour emboli, as well as micrometastatic foci (Supplementary Fig. 17). Although we cannot completely exclude the possibility of the latter hypothesis, these data supported that senescent tumour cells might move from the primary tumour mass to regional lymph nodes via lymphovascular circulation together with non-senescent tumour cells.

Senescent cells in tumour emboli of lymphovascular channels and in metastatic lymph nodes expressed CXCL12. These findings provided another important point that senescent cells could be involved in or contributed to the survival of cancer cells in the lymphovascular system. Some studies suggested that CXCL12 might be involved in tumour cell growth and survival in several types of cancers[47,48]. CXCL12 increases anoikis in colon cancer[49], whereas CXCR4 activation by CXCL12 in breast cancer cells decreases their sensitivity to anoikis[21]. Here, senescent cells enhanced anoikis resistance in thyroid cancer cell lines via CXCL12/CXCR4 signalling. Taken together, these data strongly indicated that cancer cells could invade into lymphovascular systems together with senescent cells, and that senescent cells might be involved in cancer cells survival, invasion, as well as metastasis.

Tumour formation in thyroid and metastatic foci was not observed when we performed orthotopic transplantation using SNU790 PTC cells. This might be due to the relatively indolent and low tumorigenic nature of SNU790 cells. Compared with SNU790 cells, HTH83 cells were more capable of developing tumour formation in the thyroid gland. However, the growth rate was high, resulting in tumour mass compression of the trachea and esophagus in mice 3 weeks after transplantation. Subsequently, some of the transplanted mice died of respiratory failure. Although the 3-week-period was not long enough to observe metastasis into regional lymph nodes, we could not maintain the mice for more than 3 weeks after orthotopic transplantation. This limitation may explain why we could identify only a small number of metastatic regional lymph nodes. Nonetheless, cancer invasion into lymphatic vessels was observed more frequently in the senescent thyrocytes co-transplanted mice than control mice. These data suggested that senescent cells were capable of promoting lymphovascular invasion and are involved in metastasis across the types of thyroid cancer.

Cancer invasion and metastasis are complex processes. To complete these processes, cancer cells must separate from the primary cancer, invade through the ECM, lymphatic and hematogenous systems, survive in the circulation, undergo extravasation into distant tissues or organs, and colonize them[50,51]. In recent years, cancer invasion has emerged as a hot spot in cancer research, and it has been suggested that tumour cells can acquire the ability to invade, resist apoptosis and disseminate through the EMT[1]. However, in the present study, few signs of EMT were detected in PTC. Our observations through morphological and immunohistochemical analyses, as well as 3D organoid assays, provided strong evidence of collective invasion in PTC. First, cancer cells at the invasion focus were not individual single cells, but maintained gland and/or papillary structures. Second, expression of the epithelial marker,

E-cadherin, remained in invasive epithelial cells, lymphatic tumour emboli and the lymph node metastatic region. Third, EMT markers, such as Twist1 and Zeb1, were not upregulated in invasive areas. Interestingly, instead of EMT, senescent cells were located at the front region of collective invasion in PTC. Senescent cells secrete various SASP, which can disrupt normal tissue structure and function and promote malignant phenotypes in nearby cells, as well as modify the extracellular microenvironment and remodel ECM components. In addition, senescent cells attracted cancer cells and increase cancer cell survival. Therefore, senescent cells in PTC are fully qualified as leader cells in collective cancer invasion.

In conclusion, our results demonstrated that senescent cells are actively involved in the progression of cancer and provided a new concept for the collective cancer invasion model. Senescent cells promoted collective cancer invasion and metastasis via expression of SASP. Our present observations predict a new strategy for cancer treatment, including interventions targeting cellular senescence and SASP, to improve patient outcome by inhibiting metastasis.

## Methods

**PTC samples from patients.** This study was approved by the Institutional Review Board of Ajou University Hospital (AJIRB-BMR-OBS-15-133). We obtained informed consents from all the patients who were included in the present study. PTC samples were obtained from patients at Ajou University Hospital after surgical resection. Fresh tumour and normal tissues were separately sampled in the representative areas by an experienced pathologist and snap frozen in liquid nitrogen immediately after resection, according to the tissue specimen regulation of Ajou University Hospital. Patients who had a past history of chemotherapy or radiation therapy before the surgery were excluded from the study.

**Isolation and culture of normal thyrocytes.** Normal thyrocytes were isolated from human thyroid tissues. Normal thyroid tissue was obtained after surgical resection. After confirmation with *in situ* smear cytology, normal thyroid tissue was washed five times with phosphate-buffered saline (PBS) and cut into small pieces, followed by treatment with 0.2% collagenase I (Worthington Biochemical, Lakewood, NJ) in HEPES buffer for 6 h at 32 °C. Cells were resuspended in complete F12K media (Gibco BRL, Bethesda, MD; thyroid stimulating hormone (TSH), 10 mU ml$^{-1}$; insulin, 0.01 mU ml$^{-1}$; hydrocortisone, 10 nM; transferrin, 0.005 mg ml$^{-1}$; somatostatin, 10 ng ml$^{-1}$; glycyl-ʟ-histidyl-lysineacetate, 10 ng ml$^{-1}$) and placed in 10 cm culture dishes. Next, we confirmed thyrocytes with thyroid specific markers including thyroglobulin and thyroid transcription factor-1. We further confirmed absence of BRAFV600E mutation by VE1 immunostaining.

**Thyroid cancer cell lines.** PTC cell line (SNU790) was purchased from Korean Cell Line Bank (KCLB, Seoul, Korea). Anaplastic thyroid carcinoma cell line (HTH83: established by Dr Nils-Erik Heldin, University of Uppsala, Sweden) was kindly provided by Dr Yoon Woo Koh (Department of Otorhinolaryngology, Yonsei University, Korea, cells authentication was performed by short tandem repeat (STR) analysis) and maintained in complete Roswell Park Memorial Institute medium (RPMI) media with 10% fetal bovine serum (FBS; Gibco BRL). Mycoplasma contamination of the cell lines used for this study was examined with Mycoplasma-PCR Detection Kit according to manufacture's instructions (#25235, iNtRON Biotechnology Inc., Seongnam, Korea).

**Orthotopic xenograft of cancer cells to nude mice.** All animal experiments were approved by the institutional animal research ethics committee at Ajou University Medical Center (Approval number: 2015-0008). An orthotopic xenograft of the thyroid cancer model was established by inoculating HTH83 cells into the lobe of thyroid gland[52]. Female BALB/c-nude mice (9-weeks-old) were anesthetized by intraperitoneal injection of ketamine (0.01 ml g$^{-1}$ of body weight). HTH83 cells ($9 \times 10^5$) and thyrocytes (normal or senescent; $1 \times 10^5$) were resuspended with 10 μl of PBS and then injected into thyroid gland with 30-Gauge needle. After 3 weeks, the mice were sacrificed and analysed tumour size, lymph node metastasis and lymphovascular invasion.

**Immunohistochemistry and immunocytochemistry.** Immunohistochemical staining was performed with primary antibodies on 4 μm-thick representative tissue sections of formalin fixed paraffin-embedded tissue section in the Benchmark XT automated immunohistochemistry stainer (Ventana Medical Systems Inc., Tucson, AZ, USA). The primary antibodies used were as follows: anti-BRAFV600E (VE1), predilution (#790-4855,Ventana Medical Systems Inc); p16[INK4A], predilution (#705-4713, Ventana Medical Systems Inc); Anti-Human

Ki67 antigen, clone MIB-1, 1:100 (M7240, Dako Denmark A/S, Glostrup, Denmark); TTF-1, clone 8G7G3/1, 1:50 (343M-95, Cell Marque, Rocklin, USA); D2-40 (Podoplanin), 1:100 (322M-15, Cell Marque); MMP1, 1:100 (GTX100534, GeneTex, Irvine, CA, USA); MMP3, 1:100 (GTX100723, GeneTex); MMP9, 1:100 (GTX100458,GeneTex); CXCR4, 1:100 (MAB172, R&D System, Minneapolis, MN, USA); CXCL12, 1:100 (MAB350, R&D System); E-cadherin, 1:100 (ab15148, Abcam, Cambridge, MA, USA); Twist1, 1;100 (ab50887, Abcam); Zeb1, 1;100 (NBP1-05987, Novus Biologicals, Littleton, CO, USA). Detection was done using the Ventana Optiview DAB Kit (Ventana Medical Systems). Double immunohistochemistry was performed with the UltraView Universal DAB Detection kit (#760-500, Ventana Medical Systems Inc) for first antibodies and then with the UltraView Universal Alkaline Phosphatase Red Detection kit (#760-501, Ventana Medical Systems Inc) for second antibodies in the Benchmark XT automated immunohistochemistry stainer (Ventana Medical Systems Inc). Immunohistochmical staining was scored by an experienced pathologist (J.H.K.). Intensity of immunostaining was scored as 0, 1, 2 or 3, which corresponded to negative, weak, moderate and strong staining intensity. Percentages of stained tumour cells at representative regions were recorded as numerical score in every 10% (0, 0%; 10, 1–10%; 20, 11–20%; 30, 21–30%; 40, 31–40%; 50, 41–50%; 60, 51–60%; 70, 61–70%; 80, 71–80%; 90, 81–90%; 100, 91–100%). An H score was then calculated by summing the products of staining intensities (0–3) and percentages of stained cells in each intensity (0–100); H scores ranged from 0 to 300. For the immunocytochemical staining, slides were washed with PBS two times and then primary antibodies were applied; E-cadherin, 1:100 (ab15148, Abcam); N-cadherin, 1:100 (ab18203, Abcam); VE1, (#790-4855, Ventana Medical Systems Inc). Slides were washed two times with PBS and incubated with appropriate conjugated secondary antibodies for 1 h at room temperature. Secondary antibodies for immunocytochemistry were as follows: Alexa Fluor 488, 1:600 (A-21206, Thermo Fisher Scientific, Waltham, MA, USA); Alexa Fluor 555, 1:600 (A-31572, Thermo Fisher Scientific). In the case of F-actin staining, slides were applied with rhodamine phalloidin, 1:100 (R415, Thermo Fisher Scientific) for 1 h and then analysed with fluorescence microscope.

**3D gel invasion assay.** Cells were embedded in a mixture of 10% collagen I (#354249; BD Biosciences) and matrigel (#354234; BD Biosciences, San Jose, CA, USA) in Transwell (0.4-µm pore size, Corning, Acton, MA, USA). Thyrocytes (Control, BRAFV600E or BRAFV600E/shCXCL12; $1 \times 10^5$) and SNU790 or HTH83 cells ($8 \times 10^4$) were seeded into the collagen containing matrigel mixture chamber with serum-free F12K (100 µl). F12K with 10% FBS (800 µl) was placed in the lower chamber as a chemoattractant. After 2 days, we used two kinds of invasion analysis methods. First, embedded gel was fixed with 10% formalin and then paraffin-embedded. Paraffin-embedded gel was sectioned and stained with hematoxylin–eosin (HE; Sigma). Second, embedded gel was sectioned without fixation and then the cells were analysed by fluorescence confocal microscope.

**Organoid culture.** We isolated cancer tissues from BRAFV600E-expressing PTC. Collected cancer tissues were washed with PBS three times and treated with 0.02% collagenase I (Worthington Biochemical) for 30 min at 32 °C. Separated multicells unit were centrifuged at 1,500 r.p.m. for 5 min and resuspended in complete F12K media. Suspension cells were counted and approximately 1–2 organoid µl⁻¹ in 100 µl of embedded organoid matrix using collagen I (#354236, BD Biosciences) were added to each well of Lab-Tek plate (#154917, Nunc, Roskilde, Denmark) and incubated at 37 °C for 96 h in completed F12K media.

**RNA-sequencing analysis.** Total RNA was extracted from cancer centre, peripheral region of PTC and normal, BRAFV600E-induced senescent thyrocytes using Macherey-Nagel RNA kit (Macherey-Nagel GmbH & Co. KG, Düren, Germany). Briefly, the sample quality was checked using Bioanalyzer RNA Chip (Agilent Technologies) and RNA-sequencing running was carried out with Nextseq 500 (Illumina, San Diego, CA, USA).

**Senescence associated β-galactosidase (SA-β-Gal) staining.** The cells or frozen tissue slides were fixed with 10% formalin for 1 min and then incubated with SA-β-Gal solution (X-gal, 1 mg ml⁻¹; citric acid/sodium phosphate, pH 5.8, 40 mM; potassium ferrocyanide, 5 mM; potassium ferricyanide, 5 mM; NaCl, 150 mM; MgCl2, 2 mM) for 12 h at 37 °C. After washing with PBS, SA-β-Gal-positive cells were then analysed under light microscopy. Intensity of SA-β-Gal staining in PTC frozen samples was graded as none, weak, moderate or strong (Fig. 1a).

**Migration and invasion assay.** Migration of the cells was assessed by using 3.5 cm culture dishes (Culture-Insert 2well, ibidi GmbH, Munich, Germany). Resuspended cells ($1 \times 10^4$) were seeded into each well, and the barrier between the well was removed after 12 h. Cells migration was analysed after 24 h incubation. Cell invasion assay was performed with Transwell (8-µm pore size, Corning). Uncoated membrane of Transwell insert was pre-coated with growth factor reduced matrigel (#356230, BD Biosciences). Cells ($5 \times 10^4$) were seeded into the upper chamber with 100 µl of serum-free RPMI or F12K. RPMI or F12K with 10% FBS (800 µl) was placed in the lower chamber as a chemoattractant. After 24 h, the cells invaded into the lower surface were fixed with 100% methanol for 1 min, and was stained with hematoxylin or eosin Y solution (Sigma). In the case of primary cancer cells migration assay, we seeded $5 \times 10^4$ isolated primary cancer cells into the upper chamber. After 24 h, membrane was stained with SA-β-Gal/eosin and then analysed the percentage of senescent cells in membrane upper and lower area, respectively.

**Anoikis assay.** Co-culture was performed with Transwell (0.4-µm pore size, Corning). $6 \times 10^4$ of thyrocytes (Control, BRAFV600E/shCon or BRAFV600E/shCXCL12) and $1.2 \times 10^5$ cancer cells were seeded in upper and poly-hydroxyethyl methacrylate (HEMA) (Poly 2-hydroxyethyl methacrylate, #P3932, Sigma, St Louis, MO, USA) coated lower chamber, respectively. Cell detachment induced apoptosis was analysed by Calcein AM/EthD-1 staining (Sigma) after 12 h incubation.

**Immunoblotting.** Cells were lysed in RIPA buffer (Tris pH 7.5, 20 mM; NaCl,150 mM; 1% Nonidet P-40; 0.5% sodium deoxycholate; EDTA,1 mM; 0.1% SDS) containing protease inhibitor cocktail (K272, Biovision, Milpitas, CA, USA) and phosphatase inhibitor cocktail (K282, Biovision). Samples were then resolved by SDS-PAGE and immunoblotted with the indicated antibodies; MMP1, 1:1,000 (GTX100534, GeneTex), MMP3, 1:1,000 (GTX100723, GeneTex), MMP9, 1:1,000 (GTX100458, GeneTex), p16$^{INK4A}$, 1:1,000 (ab54210, Abcam), p-Erk1/2, 1:1,000 (9106, Cell Signaling Technology, Danvers, MA), Erk1/2, 1: 1,500 (9107, Cell Signaling Technology), Bcl-2, 1:1,000 (4223, Cell Signaling Technology), Bcl-xL, 1:1,000 (2764, Cell Signaling Technology), Bax, 1:1,000 (2774, Cell Signaling Technology), CXCR4, 1:1,000 (ab124824, Abcam), BRAF, 1:1,500 (sc-9002, Santa Cruz Biotechnology, Santa Cruz, TX, USA), p-Akt$^{Ser473}$, 1:1,000 (3787, Cell Signaling Technology), Akt, 1:1,000 (9272, Cell Signaling Technology) and Tubulin, 1:2,000 (sc-32293, Santa Cruz Biotechnology). Uncropped scans of the critical immunoblots are shown in Supplementary Figs 18 and 19.

**Analysis of BRAFV600E mutation.** We used two kinds of mutant analysis methods; direct sequencing and immunohistochemistry. For genomic DNA isolation, one representative formalin-fixed paraffin-embedded tissue block of surgical specimens was selected and cut at 10-µm thickness. Genomic DNA was extracted from manually micro dissected tumour area of each tissue section using a QIAamp DNA FFPE Tissue Kit (Qiagen, Hilden, Germany) according to the manufacturer's instruction. BRAF exon 15 was amplified by PCR using the forward primer 5′-GCTTGCTCTGATAGGAAAATGAG-3′ and reverse primer 5′-GTAACTCAGCAGCATCTCAGG-3′. PCR was performed at 94 °C for 30 s, 60 °C for 30 s and 72 °C for 30 s for 34 cycles. After amplified products were purified, direct DNA sequencing was performed by Applied Biosystems 3500XL Genetic Analyzer with GeneMapper Software v4.1 (Applied Biosystems, Foster City, CA, USA). In addition to molecular analysis, we also performed immunohistochemical analysis using a mutation-specific monoclonal antibody, VE1, which was developed by Capper[53].

**Preparation of BRAFV600E and CXCR4 lentivirus.** BRAFV600E mutant and CXCR4 were cloned from normal thyroid and PTC in our laboratory[16]. cDNAs were inserted into the pCDH-CMV-MCS-EF1-Puro lentivirus vector (System Biosciences, Mountain View, CA, USA). To generate lentiviral particles, HEK-293TN cells were transfected with plasmid DNA (pGagpol, pVSV-G and pCDH-BRAFV600E or pCDH-CXCR4). For knockdown of BRAF and CXCL12 expression, shRNA was prepared in a pLKO lentiviral vector (Sigma) and then amplified in 293TN cells. Thyrocytes were plated and grown in 6 cm culture dishes. After overnight culture, they were infected with lentivirus and then the cells were selected with 3.5 µM puromycin for 1 week. shRNA sequences were as follows: shBRAF: 5′-TTACCTGGCTCACTAACTAAC-3′; shCXCL12-1: 5′-CGCCAA CGTCAAGCATCTCAAA-3′; shCXCL12-2: 5′-ACATCTCAAAATTCTCAAC ACA-3′, respectively.

**Real-time PCR analysis.** First-strand cDNA was synthesized by reverse transcription reaction using oligo-dT primers from 1 µg of total cellular RNA (Thermo Fisher Scientific). Real-time PCR was carried out with Power SYBR Green PCR Master Mix (Bio-Rad, Hercules, CA, USA) using the following conditions: initial activation at 95 °C for 5 min, followed by 40 cycles of 95 °C for 15 s and 60 °C for 1 min. The primers used for real-time PCR were given in Supplementary Table 6.

**ELISA analysis.** In total, $5 \times 10^5$ cells of control, BRAFV600E or BRAFV600E/shCXCL12 lentivirus infected thyrocyte were seeded and maintained in 2 ml of F12K media. After 48 h, the media was harvested and analysed secreted CXCL12 protein using human CXCL12 ELISA kit (R&D System).

**Caspase 3 activity assay.** We collected cells from suspended culture media and analysed caspase 3 activity according to manufacture's instructions (Promega, Madison, WI, USA).

**Statistical analysis.** Statistical analysis on SA-β-Gal staining in different areas of PTC frozen samples was performed by $\chi^2$ test. Analyses on $H$ score of immunohistochemical staining between cancer centre and invasive borders were performed using Wilcoxon signed rank test. Analyses of pathologic data from *in vivo* mice models were performed by Mann–Whitney $U$-test. Other data are presented as mean ± s.d. of independent determinations, and were analysed using Student's $t$-test, and a $P$ value < 0.05 was considered as significant. IBM SPSS ver. 22 (IBM Corp., Armonk, NY, USA) was used for all statistical analysis.

**Microscope image acquisition.** For image acquisition of histology and immuno-histochemical staining, we used ScanScope CS system (Aperio Technologies, Inc., Vista, CA, USA). The cell images were acquired using an Olympus microscope mounted with an Olympus DP70 digital camera and DP-Manager software (Olympus Microscope Corp., Tokyo, Japan) at room temperature. Immuno-fluorescence images were collected on a Zeiss LSM 510 microscope and analysed with Zeiss Axio Imager software (Carl Zeiss, Jena, Germany) at room temperature.

**Data availability.** The Raw RNA-sequencing data have been deposited in the Gene Expression Omnibus database under the accession code GSE81144 and GSE83560. The RNA-sequencing data referenced during the study are available in a public repository from the https://www.ncbi.nlm.nih.gov/geo/. The authors declare that all the other data supporting the findings of this study are available within the article and its Supplementary Information files and from the corresponding author upon reasonable request.

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

## Acknowledgements

We would like to sincerely thank Prof Seung Soo Sheen (Section of Clinical Epidemiology and Biostatistics in Clinical Trial Center and Lung Cancer Center, Ajou University School of Medicine, Korea) for his critical evaluation on statistical analysis. This experiment was supported by National Research Foundation of Korea to Tae Jun Park at Ajou University (NRF-2012R1A5A2051426, 2014R1A1A2058616), and to Jang-Hee Kim at Ajou University (NRF-2016R1D1A1B02010452).

## Author contributions

Design and contributed to analysis and interpretation of data; Y.H.K., Y.W.C., J.-H.K., T.J.P. Maintains cell and experiments including animals experiment; Y.H.K. and T.J.P. Provided patients samples and performed immunohistochemistry; J.L., E.Y.S., J.-H.K. Writing manuscript; Y.H.K., J.-H.K. and T.J.P.

## Additional information

**Competing interests:** The authors declare no competing financial interests.

**Publisher's note**: 

