## [Peer Review File · Nature Communications]

Reviewers' comments:

Reviewer #1 (Remarks to the Author):

Review of Kim et al.

Human PTC show bgal staining at invasive borders, together with p16 and MIB1. A Braf transgenic model also showed p16 and bgal staining at invasive borders. Interestingly, there did not appear to be EMT at invasive borders since Ecadherin remained high while Twist1 and Zeb1 were uniform. In the Braf model, invasive cells in a transwell assay showed beta gal staining, and SASP markers such as IL6, IL8 and MMPs were upregulated, especially at tumor borders. KD of Braf reduces bgal staining while expression of Braf in primary thymocytes induced SASP. Bgal positive cells were also present in lymphatics and lymph node metastases. CXCL1,2,12 and CXCR4 expression was increased in invasive areas compared to the tumor center. Varying CXCL12 and CXCR4 expression can affect in vitro motility and invasion properties. Coculture of tumor cells with Braf transduced thymocytes led to increased anoikis resistance which was partly dependent upon CXCL12 expression.

Overall, this paper provides interesting information regarding the potential role of senescence in stimulating collective invasion and resistance to anoikis. In addition a role for CXCL12/CXCR4 signaling in this process is proposed. The one major weakness which needs to be addressed is to take advantage of the various CXCR4 (and possibly CXCR7) inhibitors to demonstrate the mechanistic role of CXCL12/CXCR4 signaling. Thus the key experiment needed to demonstrate the role of CXCR4 signaling in the Braf transgenic model is to use the experiment of Figure 4a and test if AMD3100 inhibits the invasion. Similarly AMD3100 should be used in the studies in Figure 5.

Other points:

Figure 2B is interesting but hard to read – corresponding means, SEMs and p values relative to normal tissues should be provided as a table.

Lines 159/160 described figure 2e are incorrect – figure 2e shows shRNA data while lines 159/160 state it shows OIS data. It seems that Figure 2e and supplemental figure 4 show similar data – is there a reason for both?

Line 161 states that Braf increases invasive ability of thymocytes but I could not find the data supporting that in Figure 2.

The supplemental figure legends are inadequate and it is unclear if the methods for those figures are provided – for example the F actin staining method for Supplemental figure 5 does not seem to be provided. In addition the interpretation of Figure 5 provided in lines 171 seemed an overstatement of what could be seen in figure 5 – this figure and line 171 could be removed.

It is stated for Figure 3C that 10 cases were studied but the results are not given – just a single example is shown. A summary of the results for the 10 cases should be provided.

For the experiments in Figure 5, please indicate the source of the tumor cells (patient, animal model, cell line, etc).

More information should be provided regarding the statistical analysis leading to the histogram in Figure 6D. Perhaps a supplemental table with the data would help clarify this.

Line 640 refers to a table – which table is that?

Catalogue numbers for all antibodies, enzymes and other reagents that can show variability should be provided.

Reviewer #2 (Remarks to the Author):

The manuscript by Kim et al., examines the role of senescent thyroid tumor cells in collective migration/invasion and metastasis in human tumors and animal models. The study is interesting because previous studies from several groups have suggested that conditioned medium from senescent cells can increase migration and metastasis of tumor cells. This study attempts to correlate this within human tumors and then develops an orthotopic model to study the mechanism by which this occurs in vivo. This study is interesting, timely and would have a broad appeal. There are, however a few issues that if addressed would strengthen the manuscript. These

are listed below. A general comment about the presentation of the figures is also listed below.

1. Figure presentation, I realize Nature Communications is web based journal but many people, including this reviewer print out papers to read and many of the histological figures are too difficult to see. The authors would greatly help with reviewer/readers by addressing this deficiency (keeping in mind when the figures are type set the figures will be even harder to see because they will be further reduced in size).
2. Figure 1A, the SABGal panel is hard to see. I higher magnification would be very helpful.
3. Figure 1B, why was MIB-1 used to stain? It might be better to use Ki67, which is a well-established proliferation marker.
4. Figure 2B, there is significant variation in the SASP expression. Were there any histological grade differences in these samples? Were there more senescent cells in one tumor versus the other? A discussion of the tumors may help to explain the variation.
5. Figure 2D, in many cases the SASP expression does not correlation with the p16INK4a+ cells in the sections. What does this mean for the model? This should be discussed.
6. Throughout the figures there is little quantification. For example, Figure 2D shows a single tumor, how representative is the association of p16 and SASP expression in the invasive front versus center of the tumor across patient samples.
7. Figure 3B, the SABGal cells look to be clustered AWAY from the non-senescent cells. Should they not be touching if they are involved in collective invasion (i.e. if the nonsenescent cells are following the senescent cells)? Please discuss and quantitate how often this is observed in the primary tumors.
8. Figure 4A, again, really hard to see what they are discussing.
9. Figure 4B, they discuss an increase in CXCL12 and CXCR4 but there is no increase in CXCR4 here?? What am I missing?
10. Figure 4C, again, is there a correlation between CXCR4 and CCL12? There are only two tumors that are high expressers for these two SASP factors, is it the same tumor that is high in both or different tumors? What does this mean???
11. They move to an in vitro system to examine how CXCR4 and CCL12 impact invasion but throughout they use over-expression of CXCR4. Why? I thought the premise was that senescence induced CXCR4 and CCL2. This is one of the biggest weaknesses of the manuscript throughout. If senescence induces these factors why do the investigators need to overexpress CXCR4?
12. Figure 5D looks at the role of senescent cells and senescent-derived CCL12 in invasion. They use a single hairpin targeting CCL12. It would be stronger if there were 1) a second hairpin and 2) a neutralizing experiment to show it is only CCL12 in the conditioned medium that is responsible for the invasion and not something secondary to CCL12 knockdown in the senescent cells.
13. The argument is that tumor cells from BRAFV600E tumors can senesce yet they mix wildtype and V600E cells in Figure 6. They should discuss how this models tumors in vivo (in other words wouldn't all tumor cells express V600E?). Also, they switch tumor lines for this work and none of the in vitro characterization is done on the new cell line. How do they know the same mechanisms are active in this second tumor line since it forms tumors and the other does not in the same mice? This is a major issue that needs to be investigated or at least justified beyond this one forms tumors.
14. Figure 6B shows that there is no difference in primary tumor growth and then they go onto argue that there is a difference in invasion (Figure 6D). However, the difference is very small and it is very hard to understand how a Student's T test would produce a p value under 0.05 with the numbers that are provided in the table in Figure 6B. A more robust description of the statistics is needed here.
15. They show that senescent tumor cells are at the front of an invasive tumor and senescent tumor cells are in metastatic sites and they conclude that the senescent cells from the primary site traveled to the metastatic site, which is certainly possible but it is also possible that nonsenescent cells moved and THEN senesced upon arrival at the metastatic site. This possibility should be discussed.

Reviewer #3 (Remarks to the Author):

COMMENTS FOR THE AUTHOR:

With this work Kim and colleagues have examined senescence-associated secretory phenotype with collective sheet migration and lymph node metastasis of BRAF-mutant thyroid cancers. Strengths of the work include the human source tissue, human thyroid cancer cells, and xenograft models employed as well as the incorporation of RNA-seq and organoid culture systems.

Weaknesses of the work rest largely on the characterization of a small subset of BRAF600E-expressing thyroid cancer. The authors make the claim that they analyzed senescence in breast, prostate, colon, thyroid, stomach and lung cancers and found senescence to be elevated in thyroid. Unfortunately, there is no data or citation to support this claim. Thus, the analyzes appear to be restricted to a very limited type of cancer, and not more broadly applicable to well-differentiated collective invasion type cancers.

1. In figure 1 authors show data for the standard marker for senescence, SA-B-gal, in representative and cumulative thyroid cancers. There is no information on how the staining intensity was quantified and or measured. There are no representative images of Non, Weak, Moderate, or Strong intensity to support the subjective quantification scheme. Rationale for p16INKA or MIB was not stated. There is no corresponding staining of SA-B-gal in non-BRAF600E mutant thyroid cancers. The authors state the E-cadherin positive BRAF-mutant cells are localized to lymphatic channels but there is no immunostaining with the appropriate marker to identify those lymphatics.

2. In figure 2 the authors show correlative data that SA-B-gal cells comprise 65% of the senescence cells, despite only being 15% of the isolated cells. After completing PCR analyses of the standard IL6, IL8 and MMPs markers of SASP, they completed an RNA Seq analysis. Interestingly, these data show pronounced upregulation of additional markers including CCL21 (3.3fold), MMP13 (5.6 fold) and ADAM12 (2.0 fold) upregulated in the invasion front.

3. In figure 4 the authors attempt to complete mechanistic cell culture experiments to support the model proposed and supported with correlative immunohistochemistry in figure 3. Using RNA Seq the authors uncover increased expression of numerous chemokines, including several with 2 – 4 fold increases in expression. Only 4 chemokine receptors were analyzed and of those only CXCR4 was found to be elevated. Increased CXCR4 expression is well-established in solid and liquid tumors. The authors then focus on CXCL12 expression but do not rationalize why this ligand, rather than CCL21 (3-fold increase) or CXCL6 (4.6 fold increase) were ignored.

4. In figure 4 panel F the authors include data comparing SNU thyroid tumors against SNU transduced with CXCR4 and find that cells over-expressing the receptor are more motile across a scratch wound migration assay. Studies examining CXCR4 or inhibiting CXCR4 in the initial invasion assay from Figure 2 or the organoid in figure 4. There is a confounding array of different migration and invasion assays used in these studies. A more thorough analyses using each assay as an overlapping complement experiment would lend strength to the analyses. Thus, addition of a CXCR4 inhibitor in the scratch wound assay and the invasion assay, or utilization of the CXCL12 knockdown cells in a scratch wound assay would solidify data interpretation. There is no transcript or ELISA data comparing CXCR4 or CXCL12 expression in the lentiviral transduced cell types. Scientific rigor calls for multiple clones of lentiviral transduced cells, in either an over-expression system or genetic knockdown experiments. The level of expression needs to be validated for each of those cells and compared against an empty vector or non-specific sequence vector control.

5. None of the CXCR4 or CXCL12 staining data in figure 5 is quantified. There is an absence of data from the CXCR4-overexpressing SNU cells on the anoikis resistance phenotype. Thus, the experiment is simply correlative and doesn't demonstrate that paracrine CXCL12 signaling from senescent thyroid cells directly signals caspase in CXCR4-expressing tumor cells

6. the authors state that SNU cells do not form tumors in mice and were therefore replaced by HTH cells. However, there is no data incorporating the HTH cells into the anoikis or migration mechanistic studies in Figure 4 and Figure 5. Further, xenograft studies with the SNU-CXCR4 or SNU-shCXCL12 cells, alone or mixed with control cells should be performed to demonstrate that the correlative immunohistochemistry and cell biology studies are biologically relevant in vivo.

Reviewer #4 (Remarks to the Author):

The Authors tried to demonstrate that senescent cancer cells have an important role on tumor invasion, more interestingly, for collective cell migration. They also showed that CSCL12/CSCR4 signaling was involved in these migration and increased anoikis resistance. New concepts were extremely important and interesting. However, there were several concerns as described below.

MAJOR CONCERNS

Instead of building story based upon hypothesis, please interpretate results more modestly. The Authors used "senescent tumor cells" frequently, but some of them were only confirmed by SA- β -Gal staining. In addition, it was not sure whether these cells were dynamical change between senescent and non-senescent status.

Figure legends for supplemental data were so short and the Authors needed to describe more details.

SPECIFIC COMMENTS

Introduction

Sentence started from line 87: The Authors previously published part of preliminary study in Neoplasia (Vol 16, no 12, pp1107-1120, 2014, current reference number 51). In this manuscripts, they also showed BRAF60E-induced cell senescence in primary thyroid cells. Since results described in this article became very important evidence to further investigate role of cell senescence in current manuscript, it was better if the Authors introduced this manuscript and built background in this study.

It was better to cite any references that previously described collective cell migration in thyroid cancer, if available.

Results and Discussion

Figure 1 and others: "Thyroid cancer cells" were not confirmed as thyroid cell-specific marker. At least at the beginning, it is better to show that the cluster of cells were thyroid, using such as thyroglobulin IHC.

Figure 1B: Please show p16 staining in tumor center area as well.

Figure 1C, text line 120: It was easier to show that the positive cells were thyroid cancer cells if thyroid-specific marker and stromal cell marker in similar slice of tissue were shown.

Figure 1F: Please show staining of normal follicle area in thyroid.

Figure 2A: It is better to have SA- β -Gal staining before and after. It is possible that migrated cells might become cell senescence after they migrated.

Figure 2C (as well as figures 2F, 4B, and 4E): "vs" was confusing word. Please change to "ratio of expression, "invasive border" over "cancer center", or describe in figure legend.

Lines 157: The Authors did not show any evidences to secret SASP. They showed only increased expression. Please clarify to use correct words.

Figure 2F: There were many differences of results shown in figure 2C and figure 2F. For example, IL6 was increased in figure 2C, but not in figure 2F. The Authors should describe more carefully the results in text.

Line 198: Please provide absolute expression levels of CCLs and CCRs. It is better if the Authors select meaning full CCL and CCR based upon the expression level at first and then compare the expression levels.

Figure 4D: CXCR4 appeared to be expressed stronger in tumor center than in invasive area. Please any comments this observation.

Figure 4F: The Authors described that BRAFV600E-induced SNU790 cells were senescent. If they showed previously, please cite the paper(s). If not, please provide more details of information to confirm cell senescent.

Figure 4G: The Authors showed increased gene expression of CXCL12. They also needed to show increased protein levels in medium.

Figures 5A and 5B: It is better to show these IHC and IF using the same sample. In addition, please confirm that they are thyroid cells.

HTh83 cells: They have HRAS mutation. It is OK to use these cells, but it was better to mention the mutation status. In addition, did they express CXCL12?

Figure 6D: There was discrepancy of numbers between text and in figure. In text, lymphatic vessels/number of cancer developing mice, and in figure, cancer (+) lymph nodes/total lymph node. It was better to write both in text to avoid confusion.

Page 5, Line 19: Life technologies are now ThermoFisher Scientific, Waltham, MA.

Page 6, Line 2: Amersham Biosciences are now GE Healthcare Life Science (Pittsburg, PA).

Cells: ATCC carries none of 8505C, HTh83, and SW1736 cells. Please add correct source. In addition, please add source of TPC1 and N-thy-ori cells. Finally, cell authentication was needed.

Page 8, Line 10: Please add address information of Nanohelix.

PCR condition: The Reviewer recommends making supplemental table for primers, including sequencing and annealing information. This may save a space to add more information for other sections.

Materials and Methods

HTh83 cells: This cell line was established by Dr. Nils-Erik Heldin, not by Dr. Yoon Woo Koh. Please indicate Dr. Heldin's name or cite his paper respectfully.

Responses to Reviewer #1:

Comment 1: Overall, this paper provides interesting information regarding the potential role of senescence in stimulating collective invasion and resistance to anoikis. In addition a role for CXCL12/CXCR4 signaling in this process is proposed. The one major weakness which needs to be addressed is to take advantage of the various CXCR4 (and possibly CXCR7) inhibitors to demonstrate the mechanistic role of CXCL12/CXCR4 signaling. Thus the key experiment needed to demonstrate the role of CXCR4 signaling in the Braf transgenic model is to use the experiment of Figure 4a and test if AMD3100 inhibits the invasion. Similarly AMD3100 should be used in the studies in Figure 5.

Response: Thank you very much for providing these suggestions. According to your recommendations, we performed additional experiments to demonstrate the role of CXCL12/CXCR4 signaling using the AMD3100 antagonist of CXCR4. When we treated cell with AMD3100, CXCL12 induced expressions of p-Erk1/2 and p-Akt^{S473} were downregulated at 1 μ M concentration. Therefore, we used 1 μ M AMD3100 in experiments (Supplementary Fig. 11b).

Next, we performed cell migration, invasion, and anoikis-resistance assays by regulating CXCL12/CXCR4 signaling. First, to inhibit CXCR4 signaling in the cell migration assay, we administered 2 kinds of shCXCL12 and AMD3100. We found that CXCL12 downregulation or CXCR4 inhibition by AMD3100, significantly diminished cancer cell migration and in both SNU790-CXCR4 and HTH83 cells (Figure 4f and 4g, Supplementary Figs. 8 and 10). Furthermore, when we administered the CXCR4 antagonist (AMD3100) to the 3D organoid culture system, invasion of primary cancer cells was markedly decreased (Supplementary Fig. 16a). This data suggested that CXCL12/CXCR4 signaling played a major role on the collective invasion. Furthermore, we found that expression of MMPs were markedly downregulated in AMD3100-treated cells (Supplementary Fig. 16b), which was consistent with a previous report (Ref. 41, 42).

We further analyzed anoikis resistance in both SNU790-CXCR4 and HTH83 cells. After inhibition of CXCL12/CXCR4 signaling, we observed that CXCL12 downregulation by shRNAs or AMD3100 treatment inhibited anoikis resistance (Fig. 5c and, 5d, Supplementary Fig 14). We hope that our additional experiments are satisfactory.

Comment 2: Figure 2B is interesting but hard to read – corresponding means, SEMs and p values relative to normal tissues should be provided as a table.

Response: Thank you for this comment. Per your suggestion, we provided corresponding means, standard deviation and *p* values relative to normal tissues as a table in the supplementary data (Supplementary Table 1). Additionally, we presented the data as dot blots to clarify Figure 2b data. We hope that our changes are satisfactory.

Comment 3. Lines 159/160 described figure 2e are incorrect – figure 2e shows shRNA data while lines 159/160 state it shows OIS data. It seems that Figure 2e and supplemental figure 4 show similar data – is there a reason for both?

Response: Figure 2e shows that BRAFV600E-induced senescence in primary thyrocytes. We infected control cells (Con) with control lentivirus, with “BRAFV600E/shCon” indicating infection with both BRAFV600E lentivirus and sh-control lentivirus, and “BRAFV600E/shBRAF” indicating infection with BRAFV600E lentivirus and sh-BRAF both. Supplement Figure 5 (revised version) describes the same experimental method but we measured the invasion ability in these cells. To avoid confusion, we changed the sentence and figure legends accordingly.

Comment 4. Line 161 states that Braf increases invasive ability of thyrocytes but I could not find the data supporting that in Figure 2.

Response: We appreciate the reviewer’s comments. There was a mistake in our original version. We added an invasion assay for the *in vitro* senescence model (Supplementary Fig. 5).

Comment 5. The supplemental figure legends are inadequate and it is unclear if the methods for those figures are provided – for example the F actin staining method for Supplemental figure 5 does not seem to be provided. In addition the interpretation of Figure 5 provided in lines 171 seemed an overstatement of what could be seen in supplement figure 5 – this figure and line 171 could be removed.

Response: Thank you very much for your valuable comment. We agree that line 171 may have overstated what could be seen. However, we feel that this experiment can aid the understanding of why we focused on the roles of chemokines in the collective invasion of PTC and is, therefore, necessary to explain the present study. Therefore, we did not remove the sentence, but revised it modestly as follows:

“First, we attempted to evaluate the mechanical guidance signals by analyzing polarization of the actin cytoskeleton in PTC, but failed to identify this polarization (supplementary Fig. 6). Because senescent cells produce abundant SASP, including chemokines, we hypothesized that chemical guidance signals are involved in the collective cell migration and invasion of PTC (Fig. 3a).”

We also changed Supplementary Figure legends and methods. We hope that our changes are satisfactory.

Comment 6. It is stated for Figure 3C that 10 cases were studied but the results are not given – just a single example is shown. A summary of the results for the 10 cases should be provided.

Response: Thank you for your suggestion. We summarized the results of SA- β -gal and p16^{INK4a} staining in 10 cases of PTC, and provided the data as supplementary tables (Supplementary Table 2).

Comment 7. For the experiments in Figure 5, please indicate the source of the tumor cells (patient, animal model, cell line, etc).

Response: Thank you for this point. We analyzed anoikis resistance in both SNU790-CXCR4 [purchased in Korean Cell Line Bank (KCLB, Seoul, Korea) and established as CXCR4 expressing cells in our laboratory] and HTH83 cells [established by Dr. Nils-Erik Heldin and kindly provided by Dr. Yoon Woo Koh (Department of Otorhinolaryngology, Yonsei University, Korea)]. We described it in manuscript. We subsequently changed the Figure 5c, 5d and 5e, and Supplementary Figures 8, 10, and 14.

Comment 8. More information should be provided regarding the statistical analysis leading to the histogram in Figure 6D. Perhaps a supplemental table with the data would help clarify this.

Response: Thank you for your suggestion. In the original manuscript, we analyzed the differences of lymphovascular invasion between two groups by Student *t*-test. However, based on your suggestion, we consulted an expert in statistics (Prof. Seung Soo Sheen, Department of Biostatistics in Clinical Trial Center, Ajou University) about the results. He responded that Student *t*-test was not appropriate for analysis of the results due to the limited numbers of lymphovascular invasion and he recommended 'Mann-Whitney U test'. For more accurate evaluation, we re-examined these cases with an additional deep serial section and found additional three foci of lymphovascular invasion in cancer/senescent cells transplanted group. However, we did not find any additional lymphovascular invasion in cancer/normal thyrocytes transplanted group. We analyzed the data by Mann-Whitney U test and presented the results as the revised version of a graph and a table describing numbers of lymphovascular invasion in each case in Fig. 6d.

Comment 9. Line 640 refers to a table – which table is that?

Response: Thank you for pointing this out to us. There was a mistake our original manuscript. We indicated the table in the figure 6d. We changed the Fig. 6d and the sentence.

Comment 10. Catalogue numbers for all antibodies, enzymes and other reagents that can show variability should be provided.

Response: Per your suggestion, we added the catalog numbers of all antibodies and reagents.

Responses to Reviewer #2:

Comment 1. Figure presentation, I realize Nature Communications is web based journal but many people, including this reviewer print out papers to read and many of the histological figures are too difficult to see. The authors would greatly help with reviewer/readers by addressing this deficiency (keeping in mind when the figures are type set the figures will be even harder to see because they will be further reduced in size).

Response: Thank you for your kind advice. Per your suggestion, we changed the figures to enhance visualization of data by presenting them as higher magnification fields. We hope that our changes are satisfactory.

Comment 2. Figure 1A, the SA- β -gal panel is hard to see. I higher magnification would be very helpful.

Response: Per your suggestion, we increased the size of the figures and added higher magnification images in Figure 1a.

Comment 3. Figure 1B, why was MIB-1 used to stain? It might be better to use Ki-67, which is a well-established proliferation marker.

Response: Thank you for this suggestion. There was a mistake in our original manuscript. We used the monoclonal mouse anti-human Ki67 antigen, clone MIB-1, antibody (Dako Denmark A/S, Glostrup, Denmark), one of most commonly used antibodies for detecting of the Ki67 antigen in formalin-fixed, paraffin-embedded tissues. In diagnostic pathology, pathologists commonly called this antibody as “MIB-1”. To avoid misinterpretation and confusion, we changed “MIB-1” to “Ki67(MIB-1)”.

Comment 4. Figure 2B, there is significant variation in the SASP expression. Were there any histological grade differences in these samples? Were there more senescent cells in one tumor versus the other? A discussion of the tumors may help to explain the variation.

Response: Thank you for this important comment. In this study, we included thyroid papillary carcinoma harboring a BRAFV600E mutation, and all cases analyzed in Figure 2b represented well-differentiated carcinomas with no differences in histologic grade. However, based on your suggestion, we re-examined the clinicopathological features of six cases from those presented in Figure 2b, three cases exhibiting high expression of SASP, and other cases exhibiting low expression of SASP. We found that three cases exhibiting high expression of SASP showed more frequent lymph node metastasis than

those with low SASP expression (numbers of metastatic lymph nodes: 3, 6, and 8 vs. 0, 0, and 1, respectively). We added this difference across individual cases in the revised manuscript.

Comment 5. Figure 2D, in many cases the SASP expression does not correlation with the p16^{INK4a+} cells in the sections. What does this mean for the model? This should be discussed.

Response: Thank you for this important comment. In Figure 2d, immunohistochemical staining results for MMPs did not precisely correlate with p16^{INK4a}-immunopositive cells, because the slides were sectioned serially. Therefore, we assessed immunostaining by combined semiquantitative scoring (H score: see the next response). H score (levels of immunoexpression) of MMPs were correlated with mRNA expression of MMPs (in Fig. 2c) and were higher in invasive areas (where p16^{INK4a}-immunopositive cells were frequently present) as compared with that observed in the cancer center. These data supported that senescent tumor cells expressed SASP. Per your suggestion, we agree that immunoexpression of MMPs was more diffuse relative to that of the p16^{INK4a}-immunopositive cells. These results may be associated with the specificity of the MMP antibodies used. To improve immunostaining results, we changed antibodies to those from different manufacturers; however, we were unable to clearly remove background staining. We also suggested that paracrine effects of SASP may be involved, because SASPs can affect neighboring non-senescent cells by paracrine effects. We added H scoring data in Figure 2d and revised the manuscript accordingly.

Comment 6. Throughout the figures there is little quantification. For example, Figure 2D shows a single tumor, how representative is the association of p16 and SASP expression in the invasive front versus center of the tumor across patient samples.

Response: Thank you for this valuable comment. Per your suggestion, we re-evaluated results of immunohistochemical staining as an H score that incorporated both quantitative and qualitative assessments. Intensity of immunostaining was scored as 0, 1, 2, or 3, which corresponded to negative, weak, moderate, and strong staining intensities. Percentages of stained tumor cells at representative regions were recorded as an numerical score in every 10% (0, 0%; 10, 1–10%; 20, 11–20%; 30, 21–30%; 40, 31–40%; 50, 41–50%; 60, 51–60%; 70, 61–70%; 80, 71–80%; 90, 81–90%; and 100, 91–100%). An H score was then calculated by summing the products of staining intensities (0–3) and percentages of stained cells in each intensity (0–100), with H-scores ranging from 0 to 300. We presented the H scores of immunohistochemical staining results in Figures 2d, 4d, and Supplementary Figure 14a, 14b.

Comment 7. Figure 3B, the SA- β -Gal cells look to be clustered AWAY from the non-senescent cells. Should they not be touching if they are involved in collective invasion (i.e. if the nonsenescent cells are following the senescent cells)? Please discuss and quantitate how often this is observed in the primary tumors.

Response: Papillary thyroid carcinoma characteristically forms complex papillary structures (in 3 D). Therefore, the papillary structures are continuously connected to each other. However, sometimes the structures were seen as disconnected in slides, because the slide sections only displayed two-dimensional structures. We re-evaluated the SA- β -Gal staining data, and found that most of the cases in Figure 1a and one case in Figure 1b showed senescent tumor cells connected with non-senescent tumor cells. Immunohistochemical data for p16^{INK4A} also showed a connection between senescent cells and non-senescent tumor cells with E-cadherin in Figure 1e. To show the connection between senescent cells and non-senescent cells more clearly, we re-drew the dotted line (senescent tumor and non-senescent tumor line) in Figure 3b.

Comment 8. Figure 4A, again, really hard to see what they are discussing.

Response: We apologize for this. In Figure 4a, we wanted to present that senescent tumor cells could lead the collective invasion using an organoid 3D culture assay. We subsequently found SA- β -Gal-positive senescent cells at the front of invasive foci. We attempted to increase the size of the figure in the revised version. We hope that our changes to this figure are satisfactory.

Comment 9. Figure 4B, they discuss an increase in CXCL12 and CXCR4 but there is no increase in CXCR4 here?? What am I missing?

Response: Thank you for your very important comment. We attempted to explain that CXCL12/CXCR4 signaling was upregulated in cancer tissues. CXCL12 was upregulated in the cancer-invasive region as compared with levels observed at the cancer center. Although CXCR4 expression was increased in cancer tissues as compared with levels observed in normal follicles, CXCR4 expression was similar relative to levels in the cancer-invasive and -center regions. We also added the immunostaining H score data in Fig. 4d. To avoid confusion, we have altered the sentence accordingly.

Comment 10. Figure 4C, again, is there a correlation between CXCR4 and CXCL12? There are only two tumors that are high expressers for these two SASP factors, is it the same tumor that is high in both or different tumors? What does this mean???

Response: Thank you very much for this comment. In Figure 4c, we examined the expression of several members of the CXCLs/CXCRs family in 13 cases of BRAFV600E-expressing PTC tissues by real-time PCR, and found that CXCL1, CXCL2, and CXCL12 were significantly up-regulated in the entire cancer area (cancer center + invasive border) as compared with levels in normal tissue, whereas among their receptors, only CXCR4 was significantly upregulated (Fig. 4c, left panel). Next, we analyzed the mRNA expression of CXCL12 and CXCR4 at the invasive border and cancer center and found that CXCL12 expression was significantly increased at the invasive border (Fig. 4c, right). We feel that these results might support our hypothesis that CXCL12/CXCR4 signaling play a role in mechanical guidance signals by resulting in gradients of expression between the cancer center and the invasive border.

Based on your question, we analyzed CXCR4 and CXCL12 expression in each tumor, drew the figure as a dotted-line figure, and found that one tumor showed high expression of both CXCL12 and CXCR4, however, other tumors expressed only one of them highly. Although these collected samples had individual variations, these data revealed that levels of expression of CXCL12 and CXCR4 were increased compared to those of normal tissue.

Comment 11. They move to an in vitro system to examine how CXCR4 and CXCL12 impact invasion but throughout they use over-expression of CXCR4. Why? I thought the premise was that senescence induced CXCR4 and CXCL12. This is one of the biggest weaknesses of the manuscript throughout. If senescence induces these factors why do the investigators need to overexpress CXCR4?

Response: Thank you for your insightful comment. As mentioned, senescent thyrocytes induced by BRAFV600E showed upregulated expression of CXCL12 and CXCR4. However, in migration and invasion assays, we used a SNU790 PTC cell line to analyze whether senescent thyrocytes could attract non-senescent tumor cells (SNU790 cells). We expected that CXCR4 was upregulated in the SNU790

cell line, however, when we analyzed CXCR4 expression using real time PCR and western blot, we observed that CXCR4 was not upregulated in these cell lines. Therefore, we overexpressed CXCR4 in SNU790 cells. In the case of HTH83 cells, CXCR4 expression was upregulated relative to levels in normal thyrocytes; therefore, we did not overexpress CXCR4 in the HTH83 cell line (Supplementary Fig. 7).

Comment 12. Figure 5D looks at the role of senescent cells and senescent-derived CXCL12 in invasion. They use a single hairpin targeting CXCL12. It would be stronger if there were 1) a second hairpin and 2) a neutralizing experiment to show it is only CXCL12 in the conditioned medium that is responsible for the invasion and not something secondary to CXCL12 knockdown in the senescent cells.

Response: Thank you very for this important comment. Figure. 5d shows an anoikis resistance assay using HEMA-coated plates. According to your suggestion, we administered a second hairpin of CXCL12 (shCXCL12-2). Additionally, we performed a neutralizing experiment using a commercially available CXCR4 antagonist (AMD3100). Although AMD3100 treatment cannot completely inhibit the effects of CXCL12, inhibition of CXCL12/CXCR4 signaling induced a reduction in an anoikis resistance in thyroid cancers. (Figs. 4f and 4g, 5c, 5d and 5e, Supplementary Figs. 8, 10 and 14). We hope that our additional experiment is satisfactory.

Comment 13. The argument is that tumor cells from BRAFV600E tumors can senesce yet they mix wildtype and V600E cells in Figure 6. They should discuss how this models tumors *in vivo* (in other words wouldn't all tumor cells express V600E?). Also, they switch tumor lines for this work and none of the *in vitro* characterization is done on the new cell line. How do they know the same mechanisms are active in this second tumor line since it forms tumors and the other does not in the same mice? This is a major issue that needs to be investigated or at least justified beyond this one forms tumors.

Response: Thank you very much for this important suggestion. We agree with your concerns regarding changes in cell lines. In this experiment, we determined whether senescence cells were actively involved in the invasion and metastasis of thyroid cancer *in vivo*. As addressed in the original manuscript, we initially performed this experiment using a SNU790 PTC cell line, as well as a CXCR4-overexpressing SNU790 cell line in this revised version. Unfortunately, SNU790 cells and CXCR4-overexpressing SNU790 cells did not form tumors in nude mice (Supplementary Figure. 15). Therefore, we changed the thyroid cancer cell line to HTH83 cancer cells (BRAF wild type). And we transplanted HTH83 cells plus BRAFV600E-expressing senescent thyrocytes (or normal thyrocytes) in nude mouse. We speculated that tumor was developed by HTH83 cells and senescent cells were involved in cancer cells invasion. Furthermore, based on your suggestions, we characterized HTH83 cells by analyzing cell

migration, invasion and anoikis resistance. Our results showed that senescent cells attracted the HTH83 cells, and inhibition of CXCL12/CXCR4 signaling induced reductions in the invasion ability of HTH83 cells (Supplementary Fig. 8 and 10). Additionally, the presence of senescent cells also decreased anoikis in HTH83 cells (Supplementary Fig. 14)

Comment 14. Figure 6B shows that there is no difference in primary tumor growth and then they go onto argue that there is a difference in invasion (Figure 6D). However, the difference is very small and it is very hard to understand how a Student's T test would produce a p value under 0.05 with the numbers that are provided in the table in Figure 6B. A more robust description of the statistics is needed here.

Response: Thank you for your comments. Figure 6b shows that there was no significant difference in the size of primary tumors; however, Figure 6d reveals lymphovascular invasion, but did not indicate invasion by primary tumors. To avoid confusion, we provided a table with details describing lymphovascular invasion. In the original manuscript, we analyzed the differences of lymphovascular invasion between two groups by Student *t*-test. However, based on your suggestion, we consulted an expert in statistics (Prof. Seung Soo Sheen, Department of Biostatistics in Clinical Trial Center in Ajou University) about the results. He responded that Student *t*-test was not appropriate for analysis of the results due to the limited numbers of lymphovascular invasion and he recommended 'Mann-Whitney U test'. For more accurate evaluation, we re-examined these cases with an additional deep serial section and found additional three foci of lymphovascular invasion in cancer/senescent cells transplanted group. However, we did not find any additional lymphovascular invasion in cancer/normal thyrocytes transplanted group. We analyzed the data by Mann-Whitney U test and presented the results as the revised version of a graph and a table describing numbers of lymphovascular invasion in each case in Fig. 6d.

Comment 15. They show that senescent tumor cells are at the front of an invasive tumor and senescent tumor cells are in metastatic sites and they conclude that the senescent cells from the primary site traveled to the metastatic site, which is certainly possible but it is also possible that nonsenescent cells moved and THEN senesced upon arrival at the metastatic site. This possibility should be discussed.

Response: Thank you for your comments. We agree with the reviewer's thinking. We found neoplastic senescent cells in metastatic foci of regional lymph nodes, which raised a question of whether senescent cells in metastatic foci were from the primary tumor together with cancer cells in the manner of collective invasion or cancer cells metastasized from the primary lesion, followed by becoming senescent cells in the lymph nodes. To answer this question, we examined lymphatic channels, the

middle part of metastasis, in PTC tissues and micro-metastatic foci in the lymph nodes and we found SA- β -Gal positive and p16^{INK4A} immunopositive neoplastic senescent cells were detected in tumor emboli as well as micro-metastatic foci (Supplementary Fig. 17). Therefore, senescent tumor cells were identified from the initial staging of invasion to metastatic foci in lymph nodes. These findings led us to infer that senescent cells from the primary site traveled to the metastatic site. Considering your suggestion, we changed the sentence as follows: “Although we cannot completely exclude the possibility of the latter hypothesis, these data supported that senescent cells might move from the primary tumor mass to regional lymph node *via* lymphovascular circulation together with cancer cells.”

Responses to Reviewer #3:

COMMENTS FOR THE AUTHOR:

With this work Kim and colleagues have examined senescence-associated secretory phenotype with collective sheet migration and lymph node metastasis of BRAF-mutant thyroid cancers. Strengths of the work include the human source tissue, human thyroid cancer cells, and xenograft models employed as well as the incorporation of RNA-seq and organoid culture systems. Weaknesses of the work rest largely on the characterization of a small subset of BRAF600E-expressing thyroid cancer. The authors make the claim that they analyzed senescence in breast, prostate, colon, thyroid, stomach and lung cancers and found senescence to be elevated in thyroid. Unfortunately, there is no data or citation to support this claim. Thus, the analyzes appear to be restricted to a very limited type of cancer, and not more broadly applicable to well-differentiated collective invasion type cancers.

Response: We appreciate your insightful comments on our paper. Thyroid cancer is the most common endocrine malignancy, and more than 80% of all thyroid cancers are papillary thyroid carcinoma. Moreover, a high proportion of papillary thyroid carcinomas (40–70%) have a BRAFV600E mutation (ref; Pathology and Genetics of Tumors of Endocrine Organs in WHO classification of Tumors, 2004). Therefore, we view BRAFV600E-expressing thyroid papillary carcinoma as not qualifying as a small subset of malignancies. Senescence in human tumors is closely associated with pre-malignant stages of tumorigenesis, such as adenoma, but is absent in corresponding malignant tumors suggesting, a role for senescence as a barrier to tumor progression (Nature Rev Cancer 2010; v10, 51–57). However, in our previous study (published in Neoplasia 2014; v16, 12, 1107-1120), we found that senescent cells were still present in thyroid papillary carcinoma. In our review of the literature, we found studies (refs 11–16) indicating the presence of senescent cells in cancers having undergone neither radiation nor chemotherapeutic treatment. In our pilot investigation involving various tumor types, including breast, colon, and stomach cancers, we were also able to identify senescent cells within cancer areas. We added a Figure describing the SA- β -Gal positive tumor cells observed in different types of cancer (Supplementary Fig. 1). We apologize that we cannot provide additional data this time, because our investigation on incidences and features of senescent tumor cells in various types of cancer is ongoing, and we have not finished an evaluation of the exact role of senescent tumor cells in each cancer.

Comment 1. In figure 1 authors show data for the standard marker for senescence, SA-B-gal, in representative and cumulative thyroid cancers. There is no information on how the staining intensity was quantified and or measured. There are no representative images of None, Weak, Moderate, or Strong intensity to support the subjective quantification scheme. Rationale for p16^{INKA} or MIB-1 was not stated. There is no corresponding staining of SA-B-gal in non-BRAF600E mutant thyroid cancers. The authors state the E-cadherin positive BRAF-mutant cells are localized to lymphatic channels but there is no immunostaining with the appropriate marker to identify those lymphatics.

Response: Thank you very much for these valuable comments. Based on your suggestions, we revised the figure as follows:

(1) Per your suggestions, we added the representative images of none, weak, moderate, and strong SA-β-Gal staining in Figure 1a. We performed statistical analysis on differences in SA-β-Gal staining in each area of frozen PTC samples using chi-square test (χ^2 test) and added *p* value to the figure.

(2) Stimuli that induce senescence are signaled through various pathways, however, many of these activated p53 and essentially all of them converged in the activation of the CDK inhibitors p16, p15, p21, and p27. Additionally, because cellular senescence is based on a stable cell cycle arrest, the absence of proliferation markers, such as Ki-67 protein or 5-BrdU incorporation, is an essential condition necessary to document senescence (Nat Rev Mol Cell Biol 2014; v 15: 482–496). Therefore, we used p16^{INK4A} which is a classic marker of OIS and a useful marker of senescence *in vivo*, and Ki67 (MIB-1). In the original manuscript, we described p16^{INKA} as a senescent biomarker, and that we performed Ki67 (MIB-1) immunostaining to examine stable cell cycle arrest. To more clearly describe our rationale for using p16^{INKA} or MIB-1, we changed sentences as follows:

“Because p16^{INK4A} is a classic marker of OIS and a useful marker of senescence *in vivo*, we performed immunostaining for p16^{INK4A} using serial snap-frozen sections of the corresponding PTC and compared the expression of p16^{INK4A} with that of SA-β-Gal. We observed that the intensity and distribution of p16^{INK4A}-immunopositive cells correlated with those of SA-β-Gal positive cells (Fig. 1b), and real-time PCR analysis also indicated an increase in p16^{INK4A} mRNA expression in PTC (Supplementary Fig. 3). Because cellular senescence is characterized by stable cell cycle arrest, we also performed Ki67 immunostaining to evaluate the proliferative activity in p16^{INK4A}-immunopositive cells and confirmed the absence of Ki67 expression in these cells (Fig. 1b).”

(3) At our institute, PTC cases harboring *wild-type* BRAF are uncommon (less than 15%). According to the editorial suggestion, we did not study *wild-type* BRAF thyroid cancers; therefore, we could not include PTC cases with *wild-type* BRAF in this study.

(4) Per your suggestion, we performed D2-40 immunostaining for identifying lymphatics and added these results to Figures 1d, and 5a.

We hope that our changes are satisfactory.

Comment 2. In figure 2 the authors show correlative data that SA-B-gal cells comprise 65% of the senescence cells, despite only being 15% of the isolated cells. After completing PCR analyses of the standard IL6, IL8 and MMPs markers of SASP, they completed an RNA Seq analysis. Interestingly, these data show pronounced upregulation of additional markers including CCL21 (3.3fold), MMP13 (5.6 fold) and ADAM12 (2.0 fold) upregulated in the invasion front.

Response: In the original version of our manuscript, we showed that SA- β -Gal cells comprised 65% of the senescence cells, despite only being 15% of the isolated cells. We performed this experiment again by measuring SA- β -Gal staining after invasion in the upper area (non-migrated) and lower area (migrated). In the revised manuscript, we showed that data indicating that prior to migration, the proportion of SA- β -Gal stained cells was 22%, however, after invasion, the remaining cells in the upper area showed less than 9.5% SA- β -Gal positive staining. Additionally, we found that migrated cells in the lower area showed 62% of SA- β -Gal positive. These data indicated that senescent cells exhibited higher migratory ability. We added an image of the SA- β -Gal stained upper area following invasion in Figure 2a and in the revised manuscript accordingly. Per your comment, we evaluated SASP-related proteases and chemokines and found increased expression of several markers at the invasion border.

Comment 3. In figure 4 the authors attempt to complete mechanistic cell culture experiments to support the model proposed and supported with correlative immunohistochemistry in figure 3. Using RNA Seq the authors uncover increased expression of numerous chemokines, including several with 2 – 4 fold increases in expression. Only 4 chemokine receptors were analyzed and of those only CXCR4 was found to be elevated. Increased CXCR4 expression is well-established in solid and liquid tumors. The authors then focus on CXCL12 expression but do not rationalize why this ligand, rather than CCL21 (3-fold increase) or CXCL6 (4.6 fold increase) were ignored.

Response: Thank you for your insightful comment. We analyzed the RNA expression ratio between the cancer center and the invasive border and found increased levels of several chemokines at the invasive

area (Fig. 4b). Although several chemokines, such as CCL21, CXCL2, CCL19, CCL4, or CXCL12, were more upregulated in the invasive border as compared with the cancer center, expression of their receptors (CCR7, CXCR2, CCR1, and CCR5) was very low in PTC except for CXCL12 receptor (Supplementary Table 3 and 4). We then investigated the expression of several members of the CXCLs/CXCRs family in 13 cases of BRAFV600E-expressing PTC tissues by real-time PCR, and found that expression of CXCL1, CXCL2, and CXCL12 was significantly upregulated in the cancer area as compared with levels in normal tissue, whereas, among receptors of these chemokines, expression of only the CXCR4 receptor was significantly up-regulated. RNA sequencing data in Figure 4e shows expression of chemokines in the *in vitro* BRAFV600E-induced senescence model. Although CXCL6 expression was increased in *in vitro* model, CXCL6 expression was very low or not observed in RNA sequencing data of PTC cases (Supplementary Table 3). Therefore, we focused on CXCL12/CXCR4 signaling.

Comment 4. In figure 4 panel F the authors include data comparing SNU thyroid tumors against SNU transduced with CXCR4 and find that cells over-expressing the receptor are more motile across a scratch wound migration assay. Studies examining CXCR4 or inhibiting CXCR4 in the initial invasion assay from Figure 2 or the organoid in figure 4. There is a confounding array of different migration and invasion assays used in these studies. A more thorough analyses using each assay as an overlapping complement experiment would lend strength to the analyses. Thus, addition of a CXCR4 inhibitor in the scratch wound assay and the invasion assay, or utilization of the CXCL12 knockdown cells in a scratch wound assay would solidify data interpretation. There is no transcript or ELISA data comparing CXCR4 or CXCL12 expression in the lentiviral transduced cell types. Scientific rigor calls for multiple clones of lentiviral transduced cells, in either an over-expression system or genetic knockdown experiments. The level of expression needs to be validated for each of those cells and compared against and empty vector or non-specific sequence vector control.

Response: Thank you for this important comment. Per your suggestion, we administered a CXCR4 antagonist (AMD3100) and knockdown CXCL12 prior to performing migration and invasion assays. We found that CXCR4 inhibition or CXCL12 knockdown inhibited cancer cell migration. We changed the Figure 4f, 4g, and Supplementary Fig. 8 and 10 accordingly. Furthermore, we performed western blot analysis and ELISA to analyze transcript products in CXCR4-overexpressing cells and CXCL12 in senescent cells, respectively. Senescent thyrocytes secreted about 5-fold higher levels of CXCL12 protein as compared with secretion measured from normal thyrocytes. We added the ELISA results to Figure 4e and western blot results associated with CXCR4 to Supplementary Figure 7.

Comment 5. None of the CXCR4 or CXCL12 staining data in figure 5 is quantified. There is an absence of data from the CXCR4-overexpressing SNU cells on the anoikis resistance phenotype. Thus, the experiment is simply correlative and doesn't demonstrate that paracrine CXCL12 signaling from senescent thyroid cells directly signals caspase in CXCR4-expressing tumor cells

Response: Thank you for this important comment. Per your suggestion, we re-evaluated results of immunohistochemical staining as an H score that incorporated both quantitative and qualitative assessments. Intensity of immunostaining was scored as 0, 1, 2, or 3, which corresponded to negative, weak, moderate, and strong staining intensities. Percentages of stained tumor cells at representative regions were recorded as a numerical score in every 10% (0, 0%; 10, 1–10%; 20, 11–20%; 30, 21–30%; 40, 31–40%; 50, 41–50%; 60, 51–60%; 70, 61–70%; 80, 71–80%; 90, 81–90%; and 100, 91–100%). An H score was then calculated by summing the products of staining intensities (0–3) and percentages of stained cells in each intensity (0–100), with H-scores ranging from 0 to 300. We added H scores of CXCL12 and CXCR4 immunohistochemical staining results to Supplementary Figure 13. We performed an additional anoikis resistance experiment by applying two kinds of shRNA and commercially available CXCR4 antagonist (AMD3100) and found that inhibition of CXCL12/CXCR4 signaling induced reductions in an anoikis resistance in thyroid cancer cell lines (Figs. 4f and 4g, 5c, 5d and 5e, Supplementary Figs. 8, 10, and 14). These data suggested that paracrine CXCL12 signaling from senescent thyroid cells directly activated caspase signaling in CXCR4-expressing tumor cells. We hope that our additional experiment is satisfactory.

Comment 6. the authors state that SNU cells do not form tumors in mice and were therefore replaced by HTH cells. However, there is no data incorporating the HTH cells into the anoikis or migration mechanistic studies in Figure 4 and Figure 5. Further, xenograft studies with the SNU-CXCR4 or SNU-shCXCL12 cells, alone or mixed with control cells should be performed to demonstrate that the correlative immunohistochemistry and cell biology studies are biologically relevant in vivo.

Response: Thank you for your valid comment. Per your suggestion, we performed migration and invasion assays and an anoikis-resistance experiment using HTH83 and SNU790-CXCR4 cells (Figs. 4 and 5). In an additional orthotopic xenograft experiment, we found that SNU790 cells did not initiate tumor formation in the thyroid of nude mice. Additionally, neither SNU790-CXCR4 nor SNU790-shCXCL12 cells formed tumors in nude mice. We added SNU790 and SNU790-CXCR4 results in Supplementary Figure 15.

SNU790

SNU790-CXCR4

SNU790-shCXCL12

Responses to Reviewer #4

The Authors tried to demonstrate that senescent cancer cells have an important role on tumor invasion, more interestingly, for collective cell migration. They also showed that CXCL12/CXCR4 signaling was involved in these migration and increased anoikis resistance.

New concepts were extremely important and interesting. However, there were several concerns as described below.

MAJOR CONCERNS

Comment 1. Instead of building story based upon hypothesis, please interpret results more modestly.

Response: Thank you very much for your kind words about our paper. Per your suggestion, we modified our Results and Discussion sections to interpret our results more modestly. We hope that our changes are satisfactory.

Comment 2. The Authors used “senescent tumor cells” frequently, but some of them were only confirmed by SA- β -Gal staining. In addition, it was not sure whether these cells were dynamical change between senescent and non-senescent status.

Response: Thank you for this comment. We used a SA- β -gal assay, because it is the single most accepted and widely used marker [Dimri et al. (1995) PNAS. 92, 9363–9367, Collado et al. (2010) Nature Rev Cancer. 10, 51–57, Munoz-Espin et al. (2014) Nature Rev Mol Cell Biol. 15, 482–496]. *In vitro*, senescent cells can be confirmed by typical morphologic changes, as well as growth arrest. Unfortunately, typical morphologic changes in senescent cells in cell culture cannot be assessed *in vivo*. Therefore, we confirmed the senescence of BRAFV600E-expressing tumor cells using p16INK4A and MIB-1 markers. Senescent cells do not undergo dynamic changes between senescent and non-senescent status, because, by definition, senescent cells are in irreversible growth-arrest status [Collado et al. (2010) Nature Rev Cancer. 10, 51–57, Munoz-Espin et al. (2014) Nature Rev Mol Cell Biol. 15, 482–496]

Comment 3. Figure legends for supplemental data were so short and the Authors needed to describe more details.

Response: Thank you for this point. We supplied additional detail to the supplementary figure legends.

SPECIFIC COMMENTS

Introduction

Comment 4. Sentence started from line 87: The Authors previously published part of preliminary study in Neoplasia (Vol 16, no 12, pp1107-1120, 2014, current reference number 51). In this manuscripts, they also showed BRAF60E-induced cell senescence in primary thyroid cells. Since results described in this article became very important evidence to further investigate role of cell senescence in current manuscript, it was better if the Authors introduced this manuscript and built background in this study.

Response: Thank you for your kind words regarding our manuscript. Per your suggestion, we inserted information pertaining to our previous study in the Introduction section. We hope that revision is satisfactory.

Comment 5. It was better to cite any references that previously described collective cell migration in thyroid cancer, if available.

Response: Thank you for your comment. Per your suggestion, we attempted to locate references associated with the collective invasion in papillary thyroid carcinoma. Unfortunately, we were unable to locate any such reports.

Results and Discussion

Comment 6. Figure 1 and others: “Thyroid cancer cells” were not confirmed as thyroid cell-specific marker. At least at the beginning, it is better to show that the cluster of cells were thyroid, using such as thyroglobulin IHC.

Response: Thank you for your valuable comment. Per your suggestion, we performed TTF-1 (Thyroid transcription factor-1) immunohistochemistry as a marker of thyroid cells. We added the data in Figure 1c and 1d.

Comment 7. Figure 1B: Please show p16 staining in tumor center area as well.

Response: Thank you for providing this comment. We added data concerning the results of p16^{INK4A} immunostaining at the cancer center to Figure 1b.

Comment 8. Figure 1C, text line 120: It was easier to show that the positive cells were thyroid cancer cells if thyroid-specific marker and stromal cell marker in similar slice of tissue were shown.

Response: Thank you for this comment. Per your suggestion, we added immunostaining of TTF-1, one of thyroid-specific markers. Thyroid-specific markers including TTF-1 and thyroglobulin, however, are limited in their ability to discriminate between thyroid cancer and normal or benign thyroid diseases. In order to demonstrate that p16^{INK4A}-immunopositive cells were neoplastic senescent cells in our present PTC series, we view that investigation of BRAFV600E expression (using the VE1 antibody, which can detect BRAFV600E protein) is more crucial because BRAFV600E mutation is only found in thyroid cancer and not detected in normal or benign thyroid disease. For stromal markers, we also performed immunostaining with vimentin, which is the most commonly used stromal marker. However, vimentin was not useful, because it is co-expressed in epithelial cells of the thyroid [Am J Surg Pathol. (1989) 13:1034–1040].

Comment 9. Figure 1F: Please show staining of normal follicle area in thyroid.

Response: The upper panel of Figure 1f included immunostaining results for Twist1, Zeb1, E-cadherin, and N-cadherin proteins in the normal follicle region.

Comment 10. Figure 2A: It is better to have SA-β-Gal staining before and after. It is possible that migrated cells might become cell senescence after they migrated.

Response: Thank you for this comment. According to your suggestion, we performed this experiment an additional time. The word “before” describes the period immediately following seeding of the cancer cells isolated from PTC. This status is same as adhesion. Therefore, we measured SA-β-Gal staining after invasion in the upper area (non-migrated) and lower area (migrated). If senescent cells exhibited higher migration ability than non-senescent cells, the percentage of SA-β-Gal-positive cells in the upper area (non-migrated) would be diminished following invasion; however, if senescent cells exhibited no or lower levels of migratory ability relative to non-senescent cells, the percentage of SA-β-Gal-positive cells in the upper area (non-migrated) would be unchanged or elevated following invasion. Our data showed that before migration, the proportion of SA-β-Gal-stained cells was 22%, but after invasion, the remaining cells in the upper area constituted 9.5% of SA-β-Gal-positive stained cells. Additionally, we found that migrated cells in the lower area constituted 62% of SA-β-Gal-positive cells. These data indicated that senescent cells exhibited higher migratory ability. We added the SA-β-Gal image associated with the upper area following invasion to Figure 2a in the revised manuscript.

Comment 11. Figure 2C (as well as figures 2F, 4B, and 4E): “vs” was confusing word. Please change to “ratio of expression, “invasive border” over “cancer center”, or describe in figure legend.

Response: Thank you very much for this comment. Per your suggestion, we have changed this accordingly.

Comment 12. Lines 157: The Authors did not show any evidences to secret SASP. They showed only increased expression. Please clarify to use correct words.

Response: Thank you for this comment. There is a mistake in our original manuscript. We have changed this accordingly.

Comment 13. Figure 2F: There were many differences of results shown in figure 2C and figure 2F. For example, IL6 was increased in figure 2C, but not in figure 2F. The Authors should describe more carefully the results in text.

Response: Thank you for making this point. We agree with your comment and have revised the sentence carefully as follows:

“We found that expression patterns of MMPs were similar to those of BRAFV600E-induced senescent thyrocytes.”

We hope that our changes are satisfactory.

Comment 14. Line 198: Please provide absolute expression levels of CCLs and CCRs. It is better if the Authors select meaning full CCL and CCR based upon the expression level at first and then compare the expression levels.

Response: Thank you very much for your insightful comment. Per your suggestion, we provided the normalized absolute values of RNA expression as fragments per kilobase of transcript per million fragments mapped (FPKM) in Supplementary Table 3. We selected meaningful CXCLs/CCRs based on their expression levels and ligand-receptor interactions (Supplementary Table 4). We revised sentences. Related RNA sequencing raw data was deposited in GEO: GSE81144 and GSE83560. We hope our changes are satisfactory.

Comment 15. Figure 4D: CXCR4 appeared to be expressed stronger in tumor center than in invasive area. Please any comments this observation.

Response: Thank you for making this point. We examined immunostaining results for CXCR4 and CXCL12 in the 13 cases of PTC. Although variations in CXCR4 immunostaining existed across the PTC cases, there were no significant differences in CXCR4 immunoeexpression between the cancer center and the invasive border. We added H scores (a combined quantitative and qualitative assessment) for CXCL12 and CXCR4 in normal tissue, at the cancer center and at the invasive border in Figure 4d.

Comment 16. Figure 4F: The Authors described that BRAFV600E-induced SNU790 cells were senescent. If they showed previously, please cite the paper(s). If not, please provide more details of information to confirm cell senescent.

Response: Figure 4f shows the results of the *in vitro* migration assay. In this experiment, we showed that BRAFV600E-induced senescent thyrocytes could attract CXCR4 expressing SNU790 PTC cells. Therefore, we did not induce senescence in SNU790 cells.

Comment 17. Figure 4G: The Authors showed increased gene expression of CXCL12. They also needed to show increased protein levels in medium.

Response: Thank you for this comment. According to your suggestion, we measured secreted CXCL12 in the medium of senescent cells and shCXCL12 cells by ELISA and added these results to Figure 4e.

Comment 18. Figures 5A and 5B: It is better to show these IHC and IF using the same sample. In addition, please confirm that they are thyroid cells.

Response: Thank you for this suggestion. We performed immunohistochemical staining again and presented the results as a figure panel in figure 5a, using the same sample. Additionally, we performed double immunostaining for TTF-1 (thyroid transcription factor-1) as a thyroid cell marker and D2-40 as a lymphatic vessel marker using a slide section (left side; first image).

Comment 19. HTH83 cells: They have HRAS mutation. It is OK to use these cells, but it was better to mention the mutation status. In addition, did they express CXCL12?

Response: Thank you for this comment. Per your suggestion, we described HRAS mutation status of the HTH83 cells. Furthermore, we analyzed CXCR4 and CXCL12 expression in these cells and found that HTH83 cells expressed CXCR4 at 15-fold higher levels as compared with normal thyrocytes, whereas CXCL12 expression increased by only 2-fold relative to levels observed in normal thyrocytes. We presented these findings in Supplementary Figure 7.

Comment 20. Figure 6D: There was discrepancy of numbers between text and in figure. In text, lymphatic vessels/number of cancer developing mice, and in figure, cancer (+) lymph nodes/total lymph node. It was better to write both in text to avoid confusion.

Response: In Figure 6d, the text in the table (upper panel) indicated the number of metastatic lymph nodes. However, the graph (lower panel) displayed the numbers of lymphovascular invasion. Therefore, they are different entities. To avoid confusion, we changed Figure 6d to more clearly reflect what is being shown.

Comment 21. Page 5, Line 19: Life technologies are now ThermoFisher Scientific, Waltham, MA.

Response: We have changed this.

Comment 22. Page 6, Line 2: Amersham Biosciences are now GE Healthcare Life Science (Pittsburg, PA).

Response: We could not find “Amersham Biosciences” which you mention it in Page 6, Line 2 (or possibly in Page 16, Line 519-520) in our original paper. However, we have ensured that “Amersham Biosciences” is not in the revised manuscript.

Comment 23. Cells: ATCC carries none of 8505C, HTh83, and SW1736 cells. Please add correct source. In addition, please add source of TPC1 and N-thy-ori cells. Finally, cell authentication was needed.

Response: We were provided with an anaplastic thyroid carcinoma cell line (HTH83; established by Dr. Nils Erik Heldin, University of Uppsala, Sweden) from Dr. Yoon Woo Koh (Department of Otorhinolaryngology, Yonsei University, Korea). Per your suggestion, we changed the sentence accordingly. However, we did not use “TPC1 and N-thy-ori cells” and did not mention it in our original manuscript. If we are mistaken in this, please inform us as to the exact location of this information and we will correct it accordingly.

Comment 24. Page 8, Line 10: Please add address information of Nanohelix.

Response: We could not find “Nanohelix” which you mentioned in Page 8, Line 10 (or possibly in Page 18, Line 597-598) in our original manuscript. If we are mistaken in this, please inform us as to the exact location of this information and we will correct it accordingly.

Comment 25. PCR condition: The Reviewer recommends making supplemental table for primers, including sequencing and annealing information. This may save a space to add more information for other sections.

Response: According to your suggestion, we have added a supplementary table describing the primers used, as well as the PCR conditions in the Material and Methods section.

Materials and Methods

Comment 26. HTH83 cells: This cell line was established by Dr. Nils-Erik Heldin not by Dr. Yoon Woo Koh. Please indicate Dr. Heldin’s name or cite his paper respectfully.

Response: Thank you for this comment. Per your suggestion, we revised the sentence as follows: Anaplastic thyroid carcinoma cell line (HTH83; established by Dr. Nils Erik Heldin, University of Uppsala, Sweden) was kindly provided by Dr. Yoon Woo Koh (Department of Otorhinolaryngology, Yonsei University, Korea).

REVIEWERS' COMMENTS:

Reviewer #1 (Remarks to the Author):

The revised manuscript has adequately addressed my concerns.

Reviewer #2 (Remarks to the Author):

In general the authors have addressed many of my concerns. There remain two points that confuse me; 1) they claim the senescent cells migrate (Fig.2) but given they have a mixed population (+/- senescence) it is quite possible it is the non senescent cells that migrate and then senesce. I asked about this last time and I still don't think compare the percent of senescence versus non senescent cells in the beginning population answers this question. If they want to really address this they need 100% senescent cells to start and ask if they migrate/invade. Short of this they need to leave the possibility that senescent cells "enable" non senescent cells to migrate/invade open. and 2) I still do not understand why they mix wild type and BRAF overexpressing cells in Fig 6, as stated before, shouldn't all the tumor cells be expressing BRAF?

Reviewer #3 (Remarks to the Author):

None.

Reviewer #4 (Remarks to the Author):

The Authors responded most of critiques. Two more comments.

1 Normal thyrocytes (paragraph starts from line 280): Please add more information of normal cells in Materials and Methods.

2 Cell Authentication: Please state that the cell lines used here were confirmed the same cells, which were originally described, by STR analysis if examined. If not, please confirm them.

REVIEWERS' COMMENTS:

Reviewer #2 (Remarks to the Author):

In general the authors have addressed many of my concerns. There remain two points that confuse me; 1) they claim the senescent cells migrate (Fig.2) but given they have a mixed population (+/- senescence) it is quite possible it is the non senescent cells that migrate and then senesce. I asked about this last time and I still don't think compare the percent of senescence versus non senescent cells in the beginning population answers this question. If they want to really address this they need 100% senescent cells to start and ask if they migrate/invade. Short of this they need to leave the possibility that senescent cells "enable" non senescent cells to migrate/invade open. and 2) I still do not understand why they mix wild type and BRAF overexpressing cells in Fig 6, as stated before, shouldn't all the tumor cells be expressing BRAF?

Response to Reviewer #2

1) Thank you for your comment. We agree with the reviewer's thinking. Per your suggestion, we changed the sentence in the results section of Fig 2 (revised version Fig. 3). and addressed your concern as follows: "Although the possibility that senescent cells enable non-senescent cells to migrate/invade and the non-senescent cells undergo senescence after migration/invasion is present, these data suggested that senescent cells exhibited a higher invasive ability than non-senescent tumor cells."

2) Thank you for this comment. As mentioned, all tumor cells in PTCs harboring a *BRAFV600E* mutation express BRAFV600E. However, in Fig.1, we showed that not all BRAFV600E-expressing tumor cells undergo senescence. Therefore, to mimic this condition of primary PTCs *in vivo*, we required a mixture of senescent cells and non-senescent thyroid cancer cells. In this experiment (revised version, Fig. 8), we initially performed cell transplantations using a mixture of senescent cells (*BRAFV600E*-induced senescent cells) and non-senescent thyroid cancer cell lines harboring a *BRAFV600E* mutation (SNU790 PTC cell line). Unfortunately, neither SNU790 cells nor CXCR4-overexpressing SNU790 cells (all harboring a *BRAFV600E* mutation) formed tumors in nude mice (Supplementary Fig. 15). We attempted to acquire other thyroid cancer cell lines harboring *BRAFV600E* mutations and capable of forming tumors in orthotopic xenograft models, but we were unsuccessful. Therefore, we changed the thyroid cancer cell line to HTH83 cancer cells (harboring wild type *BRAF* and exhibiting higher *CXCR4* expression). In this experiment, we focused on the role of senescent cells in the primary tumor in invasion and metastasis rather than the role of the *BRAFV600E* mutation. Furthermore, based on your previous suggestions, we characterized the HTH83 cells by analyzing cell migration, invasion and anoikis resistance. We hope that our explanations are satisfactory.

Reviewer #4 (Remarks to the Author):

The Authors responded most of critiques. Two more comments.

Comment 1. Normal thyrocytes (paragraph starts from line 280): Please add more information of normal cells in Materials and Methods.

Response: Thank you for this comment. We isolated normal thyrocytes from the normal areas (non-cancerous) of fresh thyroid tissue immediately following surgical resection of *BRAFV600E* harboring PTCs. After isolation of the primary thyrocytes, we confirmed thyrocytes with thyroid specific markers, including thyroglobulin and thyroid transcription factor-1. We further confirmed the absence of the *BRAFV600E* mutation by VE1 immunostaining. We described the methods associated with isolation of normal thyrocytes in detail in the Methods section.

Comment 2. Cell Authentication: Please state that the cell lines used here were confirmed the same cells, which were originally described, by STR analysis if examined. If not, please confirm them.

Response: Thank you very much for providing these suggestions. According to your recommendations, we performed STR analysis to clarify the status of HTH83 cells. The results matched exactly those of previous reported data (*Clin Cancer Res*, 2011, 17(8):2281-2291, markers : CSF1PO 11; D13S317 11, 13; D16S539 11, 12; D5S818 12; D7S820 12; TH01 6, 9; TPOX 8; vWA 19). We described it in Methods section.

20170215-str